# Melatonin and Coenzyme Q10 mitigate Senescence in Human Adipose-Derived Mesenchymal Stem Cells by Restoring Mitophagy and Mitochondrial Proteostasis

**Aleena Vikraman, Logeswari Ravi, Naveena Kandasamy, Anuradha Dhanasekaran**[ID]*

Department of Biotechnology, Anna University, Chennai, India

* anu@annauniv.edu

## Abstract

Mitochondrial quality control is a crucial factor governing self-renewal capacity, maintenance of metabolic balance, and cellular longevity in stem cells. Impaired mitophagy significantly contributes to cellular senescence, causing accumulation of damaged mitochondria and impaired proliferative capacity of cells, leading to reduced therapeutic efficiency. This study explores mitophagy's role in regulating senescence in human adipose-derived mesenchymal stem cells (HADMSCs) and evaluates the therapeutic potentiality of antioxidants—melatonin and coenzyme Q10 (CoQ10) targeting mitochondria. It also examines the impact of antioxidant intervention aimed at improving the fate and survival, thereby establishing a connection between metabolic reprogramming and mitophagy. Our study found that stress-induced HADMSCs have reduced Mitochondrial Membrane potential (MMP), increased ROS, and increased senescence-associated β-galactosidase activity as observed through fluorescence-based imaging and biochemical assays. It was observed that antioxidant intervention has prevented the damage caused by the stress and reduced mitochondrial ROS and lipid peroxidation and has significantly restored mitophagy markers like Parkin, NDP52, BNIP3, BNIP3L/Nix, and LC3B. Our findings suggest that antioxidants induced pharmacological stimulation of mitophagy could potentially reverse stem cell aging and prevent functional decline, thereby improving regeneration and offering new insights and perspectives on mitochondrial health for improved efficiency of stem cell transplantation, maintenance and longevity of HADMSCs.

## 1. Introduction

Mesenchymal stem cells (MSCs) have shown promising potential for stem cell therapy because they can self-renew, differentiate into multiple lineages, and improve metabolic homeostasis, making them a potential candidate for regenerative therapy [1]. MSCs undergo progressive aging through Reactive Oxygen Species (ROS)

**Data availability statement:** All relevant data are within the paper and its Supporting information files.

**Funding:** "This work was supported by the Research Fellowship provided by University Grants Commission (UGC), Government of India, through National Eligibility Test (NET Fellowship- JRF & SRF awarded to Aleena Vikraman [(Award letter: F.No.16-6(DEC 2018)/2019(NET/CSIR); UGC Ref. no. 470/ (CSIR-UGC NET DEC 2018)" "The funders had no role in study design, data collection and analysis, decision to publish, or preparation of the manuscript".

**Competing interests:** The authors have declared that no competing interests exist.

accumulation and replicative senescence, causing degenerative changes and oxidative stress leading to DNA damage, telomere shortening, protein damage, and mitochondrial dysfunction, undergoing morphological and functional alterations causing senescence in MSCs [2].

Senescence affects the behavior of stem cells and their surroundings by secreting factors that can alter non-senescent cells to a hyper secretory phenotype called Senescence Associated Secretory Phenotype (SASP), containing pro-inflammatory cytokines and proteases, leading to metabolic disorders, neurodegenerative diseases like Alzheimer's, and other age-related diseases, which are maladaptations of prolonged exposure of healthy cells to senescent cells [3]. Thus, reversing senescence becomes essential in these cells to restore normal cell function and improve healthspan. Although believed irreversible, senescent cells can re-enter the cell cycle under certain circumstances which states that aging is a quasi-program [4].

Transplanted MSCs show low survival and engraftment rates due to inflammation and damage caused by free radicals at the site of injury [5].Oxidative stress, destroying biological activity may contribute to the poor therapeutic efficiency of transplantation [6]. Premature senescence in MSCs caused by ROS requires both non-enzymatic and enzymatic antioxidant systems to protect cells from oxidative stress [7].

In stem cells excessive free radical production and its accumulation leads to oxidative stress, loss of stemness, and stem cell dysfunction; hence, redox homeostasis is crucial for maintenance and long-term *in-vitro* expansion [8]. During *in-vitro* proliferation for clinical applications, cellular redox homeostasis, supported by antioxidant systems, becomes essential for preserving stem cell activities and preventing oxidative stress induced senescence [8,9]. Thus, it is essential to investigate various approaches to enhance health and longevity of stem cells.

Mitophagy is a crucial autophagic process that eliminates dysfunctional mitochondria by lysosomes, ensuring cellular health and maintaining precise regulation of cellular homeostasis [10]. High ROS released from mitochondria exacerbate cellular damage, as it not only damages mitochondria but also impair proper regulation of mitophagy; damaged mitochondria activate inflammatory signals, causing further damage, and accumulated dysfunctional mitochondria give constitutive signals, forming a loop of signals-damage response [2]. This process is crucial for health and longevity, particularly in shielding stem cells from metabolic stress, as impaired mitophagy leads to accumulation of impaired mitochondria and exacerbates aging [2]. Epigenetics and mitophagy are interconnected processes, with mitochondrial function influencing and being influenced by epigenetic modifications, creating a feedback loop resulting in the determination of fate of the stem cells [11]. So, curbing the excessive ROS in the system becomes essential to ensure efficient clearance of damaged mitochondria through regulated mitophagy, where we hypothesize antioxidant supplementation could become beneficial.

Antioxidants directly scavenge ROS and are also found to enhance the endogenous antioxidants system within the cells, which declines progressively with age [12]. Melatonin & CoQ10 are potential antioxidants that are synthesized naturally and also can be administered as supplements [13,14]. The study focuses on how proteins

involved in mitophagy are directly impacted by melatonin & CoQ10 in HADMSCs. Understanding the synergistic effects of the role of antioxidants and the mitophagy signaling pathway could lead to novel strategies of antioxidant interventions to delay stem cell aging, alleviate oxidative damage and ROS-induced senescence, and improve the efficacy of stem cell regenerative therapy.

Cellular reprogramming in stem cells can be achieved by chemical reprogramming or epigenetic modifications, where antioxidants could directly or indirectly contribute to reprogramming to mitigate senescence [15]. Even though co-enzyme Q10 & melatonin cannot be grouped under senolytic drugs, they are observed to have broader capabilities reducing the incidence of cellular senescence. The role of these antioxidants in cellular reprogramming through mitophagy is yet to be fully explored.

Antioxidant supplementation in stem cell therapy is a paradox, as it can scavenge ROS but may act as pro-oxidants, requiring further investigation [16]. Preconditioning MSCs with low $H_2O_2$ levels provide cyto-protection while maintaining physiological functions [5]. This underscores the necessity for precise modulation of stressors, as even mild oxidative challenges require meticulous calibration. Our study systematically optimized antioxidant and stress factor concentrations, tailoring strategies like antioxidant pretreatment and post-treatment to enhance therapeutic efficacy while safeguarding cellular integrity. These studies will substantially advance our understanding of oxidative stress management in stem cell cultures, thereby enabling the development of optimized protocols that enhance stem cell viability, functional longevity and therapeutic potency, which are critical determinants for the successful implementation of stem cell-based regenerative therapies.

## 2. Materials and methods

### 2.1 Cell culture of HADMSCs

In this work we used human adipose-derived mesenchymal stem cells (HADMSCs) procured from Hi-Media as a kit (EZXpand™ Adipose Mesenchymal Stem Cell Culture Kit—Cat. No. CCK024). Cells were grown and sub-cultured in tissue culture-treated T-25 cell culture flasks (NEST, Cat. No. 707003) with HiMesoXL™ Mesenchymal Stem Cell Medium containing 10% fetal bovine serum (FBS) and 1% penicillin/streptomycin (Pen/Strep) maintained at 37°C and 5% $CO_2$ incubator. Media was changed every 2–3 days and was sub-cultured when the cells reached 70–80% confluency with trypsin EDTA.

### 2.2 Treatment of HADMSCs

HADMSCs of Passage 7 & 8 were used for the current study. There were six sample sets included to study different cellular micro-environments, which included control cells (HADMSCs alone), cells treated with stress factors (HADMSCs + D-Gal + $H_2O_2$) (100μM $H_2O_2$ and 100mM D-galactose), cells treated with antioxidants melatonin (HADMSCs + Mel) and coenzyme Q10 (HADMSCs + CoQ10) alone, coenzyme Q10 pretreated cells prior to induction of stress factors (HADMSCs + coenzyme Q10 + D-Gal + $H_2O_2$), and melatonin pretreated cells (HADMSCs + melatonin + D-Gal + $H_2O_2$). The antioxidant treatments were given 24 hours after seeding cells, and stress factors were given after 6 hours of antioxidant treatment.

### 2.3 Evaluation of cell viability by MTT assay

The viability of HADMSCs was assessed by MTT (3-(4,5-dimethylthiazol-2-yl)-2,5-diphenyltetrazolium bromide) assay [17], which measures the viable cell population and reflects the overall metabolic state of the cells with regard to their mitochondrial activity. 5 x $10^3$ cells per well were seeded in a 96-well plate. 72 hours after treatment, the media was removed and washed with PBS, and 100μl of MTT (0.5 mg/mL) reagent in serum-free media was added to each well and incubated for 4 hours in the dark in an incubator at 37°C with a 5% $CO_2$ supply. The reagent was then carefully removed,

and 100 µl of DMSO was added to dissolve the formazan crystals, and after 15 minutes the absorption was measured using a spectrophotometer at 570nm.

## 2.4 Senescence-associated β-galactosidase assay (SA β-Gal)

The HADMSCs were seeded in a 24-well plate and were allowed to reach 70% confluency. 72 hours after the treatment, the media was removed and cells were washed with 1x PBS, then fixed with 1% paraformaldehyde for 10 minutes at room temperature. The fixed cells were rinsed twice with 1x PBS. SA-β-Gal staining solution containing: 1 mg/mL X-Gal in dimethylformamide (DMF), 5 mM potassium ferricyanide, 5 mM potassium ferrocyanide, 150mM NaCl, and 2mM $MgCl_2$ in 40mM citric acid/sodium phosphate buffer (pH 6.0) was freshly prepared. 250–300 µL of staining solution per well were added and incubated at 37°C without $CO_2$ exposure in the dark overnight [18]. The senescent cells had blue-colored depositions when observed under a bright-field microscope, imaged, and quantified by ImageJ software for SA-β-Gal-positive cells.

## 2.5 Detection of ROS by DCFH-DA staining

The cells after treatment were washed with PBS and then incubated with the staining solution containing 5µM of the dye DCFH-DA (2′,7′-dichlorodihydrofluorescein diacetate) prepared in serum-free media and incubated in the dark at 37°C for 30 mins. Then the fluorescence intensity at 485nm/527nm excitation and emission, respectively, was measured for triplicate samples on a 96-well plate (spark plug device) spectrophotometer [19], which was repeated thrice. Microscopic images were taken using an epi-fluorescence microscope and were analyzed for fluorescence intensity with ImageJ software [20].

## 2.6 Antioxidant assays

**2.6.1 Total Antioxidant Capacity (TAC) by ABTS.** The total antioxidant capacity of the cellular system was measured by ABTS+•-scavenging [21]. The ABTS+• radical was produced by mixing and incubating equal volumes of 7.4mM ABTS with equal volumes of 2.45mM potassium persulfate for 12–16 hours in the dark. This solution is further diluted with distilled $H_2O$ to achieve an OD of 0.7 to 1.0 when measured at 734 nm. To the 10µl of cell lysate, 200µl of the diluted ABTS+• radical solution was added, and the absorbance was measured after 6 minutes using a plate reader [22].

**2.6.2 DPPH antioxidant assay.** To 500µl of conditioned media, 0.03mM DPPH (2, 2-diphenyl-1-picrylhydrazyl) solution is added. The DPPH is initially dissolved in ethanol and then diluted to a working solution with PBS. It is then incubated at room temperature for 1 hour, and the absorbance is read at 532nm in a microplate reader. The DPPH radical scavenging percentage was calculated by the formula (absorbance of blank − absorbance of treated cells/absorbance of blank) ×100 [22].

**2.6.3 Superoxide Dismutase (SOD) assay.** The treated cells were lysed using a lysis buffer, and the protein content was measured using Bradford reagent to estimate the total available protein for the sample with which the SOD values are normalized for each sample. The assay was performed by the pyrogallol method, and superoxide anion radicals were measured using the formula mentioned in [23] and normalized with the protein concentration. Triplicate samples were used, and the assay was performed thrice.

**2.6.4 Catalase (CAT) assay.** The cell lysates of each sample were estimated for their protein concentration, and the catalase assay [24] was performed in triplicate for each sample. They were performed thrice, and the catalase activity was measured and normalized with the protein concentration and statistically analyzed for significance.

**2.6.5 Micro $H_2O_2$ assay.** 100µl of cell lysate was incubated with 50µl of $KMnO_4$ and 1M $H_2SO_4$ for 10 minutes in the dark by shaking, and the absorbance was measured at 550–652nm. The $H_2O_2$ contents in the samples were measured, plotted into a graph, and compared with a standard $H_2O_2$ solution [25].

 

**2.6.6 Glutathione (GSH) assay.** The GSH in the cell lysate, which is a natural antioxidant tripeptide in stem cells, is essential to maintain redox balance by scavenging ROS. It was measured by colorimetry using DTNB, known as Ellman's reagent [26]. The GSH content in the lysate was normalized with the protein concentration and compared with a standard GSH solution [27].

**2.6.7 Glutathione S-transferase (GST) assay.** Glutathione-S-Transferase is an enzyme that protects cells from various damage and acts as a catalyst in the detoxification process. GST activity was assayed spectrophotometrically at 25°C with reduced glutathione (GSH) and 1-chloro-2,4-dinitrobenzene (CDNB) as substrates [28]. The protein lysate was incubated in PBS, 10mM GSH, and 60mM CDNB, and the difference in absorbance was immediately measured at 340nm every minute for 5 minutes. GST activity was measured using the formula mentioned in [24].

**2.6.8 Lipid peroxidation assay.** The lipid peroxidation was estimated by measuring the amount of TBARS (Thio-Barbituric Acid-Reactive Substances) in the sample. The cell lysate was precipitated with 10% TCA, and the supernatant collected was treated with an equal amount of TBA, which was then heated at 100°C for 10 minutes. After cooling, the absorbance of the samples was measured at 532nm [29].

## 2.7 Mitochondrial Membrane Potential (MMP) analysis by JC-1 staining

The mitochondrial membrane potential was detected by JC-1 stain. After seeding the cells and treating them with antioxidants and stress factors, the media was removed and washed with PBS, then 5µM JC-1 dye solubilized in 50µl DMSO was mixed with serum-free media, which was then added to cells and incubated at 37°C for 30 minutes. Then the stain was removed and washed with PBS thrice, imaged under a fluorescence microscope and analyzed for the ratio of red and green fluorescence with ImageJ software [30]. The JC-1 probe has dual-emission fluorescent dye that directly correlates with the difference in mitochondrial membrane potential (MMP). JC-1 appears as red aggregates in healthy mitochondria and forms green monomers in depolarized mitochondria, which are detected by excitation/emission of 540/570nm and 485/535nm respectively. The results were represented as a ratio of red to green fluorescence compared with the control [31].

## 2.8 Acidic vacuole staining by acridine orange

The metachromatic fluorophore acridine orange (AO) (0.1 mg/mL) was used to measure lysosomal activity. Following treatment, HADMSCs underwent two gentle PBS rinses to remove debris before being exposed to acridine orange staining for one minute, and then the excess stain was removed by PBS. An inverted fluorescence microscope was used to view the fluorescence after the cells [32].

## 2.9 Western blot

The treated cells were lysed with RIPA buffer [33] containing protease and phosphatase inhibitors and centrifuged at 12,000 rpm for 30 minutes at 4°C. The supernatant was collected and estimated for protein concentration using Bradford reagent. The lysates were mixed with Sample Solubilization Buffer (SSB) and boiled for 10 mins at 95°C. Equal concentrations of each sample were then loaded into each well of SDS-PAGE and resolved. The resolved gel was then transferred to a methanol-activated PVDF membrane and blocked with 5% skimmed milk for 1 hour, washed, and incubated with primary monoclonal antibody (mAb) for 4 hours or overnight, then again washed and incubated with the corresponding secondary antibody for 1 hour, and imaging was done in Bio-Rad ChemiDoc. The bands were imaged, and expression levels were quantified using Image Lab software. [Primary Antibodies (Cell Signaling Technology): Parkin (Prk8) Mouse mAb #4211, Optineurin (D2L8S) Rabbit mAb #58981, NDP52 (D1E4A) Rabbit mAb#60732, BNIP3 (D7U1T) Rabbit mAb #44060, BNIP3L/Nix (D4R4B) Rabbit mAb #12396, LC3B (D11) Rabbit mAb #3868, beta-Actin (8H10D10) Mouse mAb #3700. Secondary Antibodies: Anti-rabbit IgG, HRP-linked Antibody #7074 (Cell Signaling Technology) and Bovine Anti-mouse IgG, HRP-linked Antibody SC-2380 (Santa Cruz Biotechnology)]

## 2.10 Data analysis

All the experiments were performed thrice with triplicate samples; data were expressed as mean with ± standard deviation (SD) and analyzed for statistical significance using GraphPad Prism (version 8.0.1, GraphPad Software, San Diego, USA). A one-way ANOVA was used to compare the experimental conditions to derive the statistical significance. Tukey's test was performed for post-hoc analysis to observe which treatments significantly differ from the other treatments. $P < 0.05$ was considered statistically significant.

## 3. Results

### 3.1 CoQ10 & melatonin have promoted cell proliferation at a concentration less than the micromolar range and also have a protective effect when stress is induced

Antioxidant treatment has shown a dose dependent effect on HADMSC proliferation and viability, where at lower concentrations antioxidants enhanced proliferation at $10^{-7}$ M for both CoQ10 & melatonin ($P < 0.0001$), whereas it is observed to elicit a cytotoxic response at $10^{-3}$ M concentration. Antioxidants promoted proliferation over a narrow concentration range from $10^{-10}$ M to $10^{-7}$ M range (Fig 1A). Consistently, it was observed that the proliferation peaked at 250nM - 300nM, corresponding to 120% − 142% increase (Fig 1B). Exposure to $H_2O_2$ for 2 hours led to a significant reduction in the quantity of viable cells, whereas 1 hour exposure & 30 minutes of exposure showed a comparatively lesser reduction in the cell viability, where100µM $H_2O_2$ exposure-maintained cell viability at around 83% ($P < 0.0001$), which was selected as an optimal condition to induce sublethal stress that directs cells towards senescence (Fig 1C). In parallel, to establish a complementary stress condition, the LD50 for D-galactose in HADMSCs was determined to be 150mM ($P < 0.0001$), where 100mM concentration was selected to induce cellular stress throughout the experiments ($P < 0.0001$) (Fig 1D). Under these stress conditions, the $H_2O_2$ (100µM) 1-hour exposure & D-Gal (100mM) treatment reduced viability to approximately 75%, providing a baseline for assessing antioxidant protection (Fig 1E). In HADMSCs treated by supplementing antioxidants alone, CoQ10 and melatonin produced maximum proliferative response at 300nM and 400nM concentrations after 72 hours post treatment, reaching about 121% and 113% viability, respectively (Fig 1E. I). In pretreatment settings where cells were exposed to antioxidants prior to stress, antioxidants were observed to have some protective role and showed the highest proliferation at 400nM concentration, achieving 117% & 98% viability for CoQ10 & melatonin, respectively (Fig 1E. ii). By contrast, in the post-treatment setting in which antioxidant treatments were given after the stress exposure, the highest viability was at 200nM concentration, which is about 100.1% & 87.3% for CoQ10 & melatonin respectively (Fig 1E.iii). Considering both treatment strategies, it was observed that 400nM concentration of antioxidant intervention showed better cell viability in the pretreated group compared to the post-treated group showing increases of 20.2% and 13.2%, respectively. These findings indicate that 400nM of CoQ10 & melatonin pretreatment has been shown to improve HADMSC proliferation more effectively than post-treatment under stress conditions (Fig 1E) ($P < 0.0001$) (Two-way ANOVA). Overall, when comparing stressed versus antioxidant-treated HADMSCs prior to stress, there is significant viability ($P < 0.05$) (Fig 1F), both CoQ10 & melatonin exert a significant protective effect even at a concentration as low as 400nM ($P < 0.0001$) (Fig 1B).

### 3.2 Pre-treatment of HADMSCs with low concentrations (400nM) of CoQ10 & melatonin prior to stress induced by $H_2O_2$ and D-gal elicits better cyto-protection and mitigated senescence than post-treatment after stress

SA-$\beta$-Gal staining demonstrated that increasing $H_2O_2$ concentrations induced senescence-like changes in HADMSCs, with cells exposed to 200 µM concentrations found to have disrupted morphology with extensive staining (Fig 2A). Quantification of $\beta$-Gal-Positive cells revealed that 100µM concentration showed the highest percentage area, indicating the strongest senescence response, which was almost a 10-fold increase in positive area relative to untreated controls ($P < 0.0001$). The percentage area declined at concentrations greater than 100µM, consistent with loss of viable cells

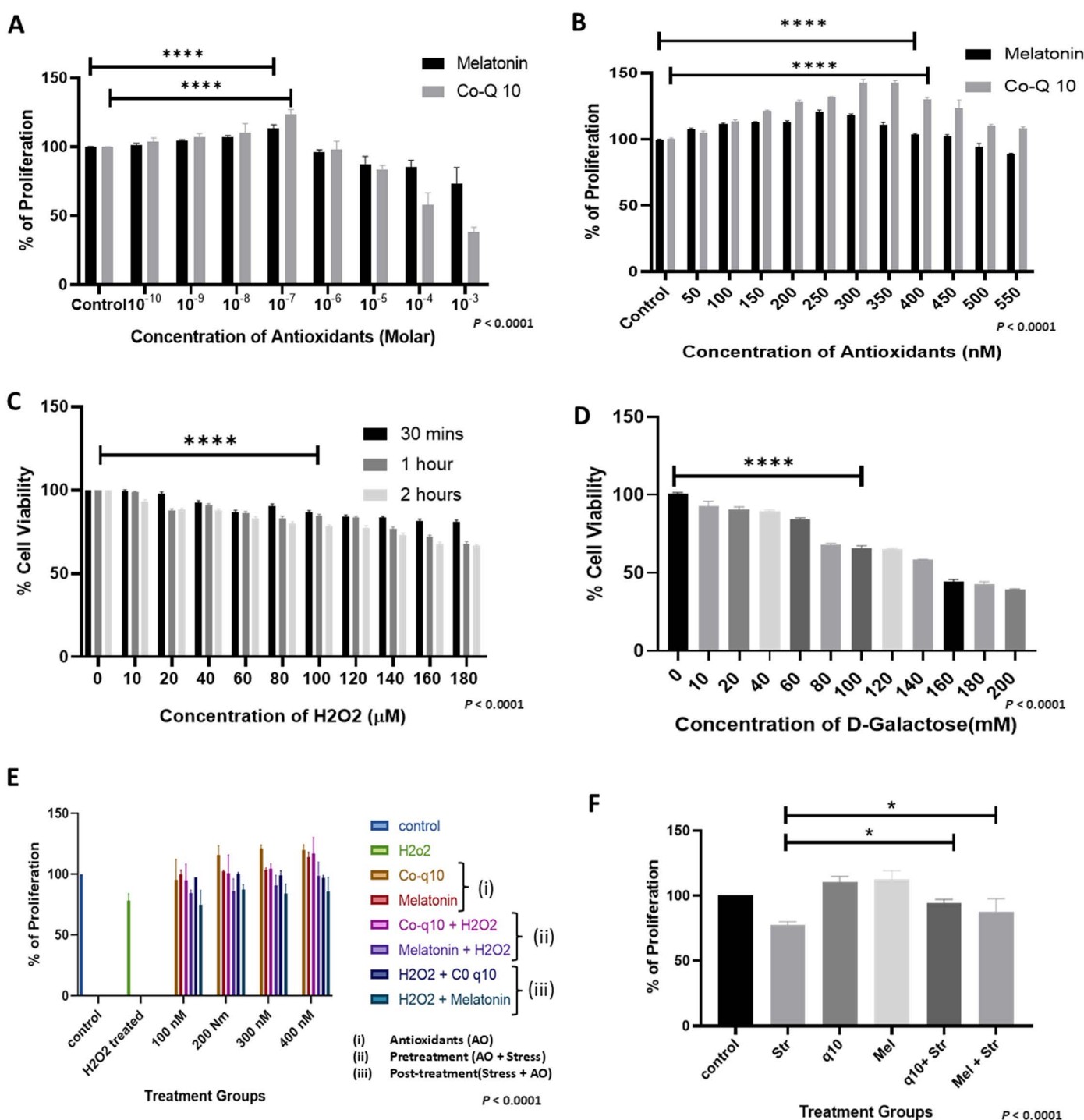

**Fig 1. Impact of antioxidants on cell viability.** MTT assay in HADMSCs: 72 hours after treatment with antioxidants (melatonin & CoQ10) across a broad range of concentrations **(A)** from $10^{-3}$ M to $10^{-10}$ M and a narrow range **(B)**. 24 hours after treatment with $H_2O_2$ **(C)** for 30 mins, 1 hr. and 2 hrs. exposure time, D-galactose **(D)**. Data are mean±SEM (n=3), normalized to control (100%) ($P < 0.0001$). **(E)** 72 hours after treatment for different treatment groups: E(i) antioxidants alone, E(ii) antioxidant pretreatment, E(iii) antioxidant post-treatment. Data are mean±SEM(n=3), normalized to control (100%) ($P < 0.0001$). **(F)** Finalized set of samples: [Control, STR ($H_2O_2$ 100μM/1 hour + D-Galactose 100mM), CoQ10 (400nM), Melatonin (400nM), CoQ10+STR, Mel+STR]—24 hours post treatment ($P < 0.0001$; n=3). All the results were subjected to one-way ANOVA. Post hoc analysis performed by Tukey's test (*=$P < 0.05$; ****=$P < 0.0001$).

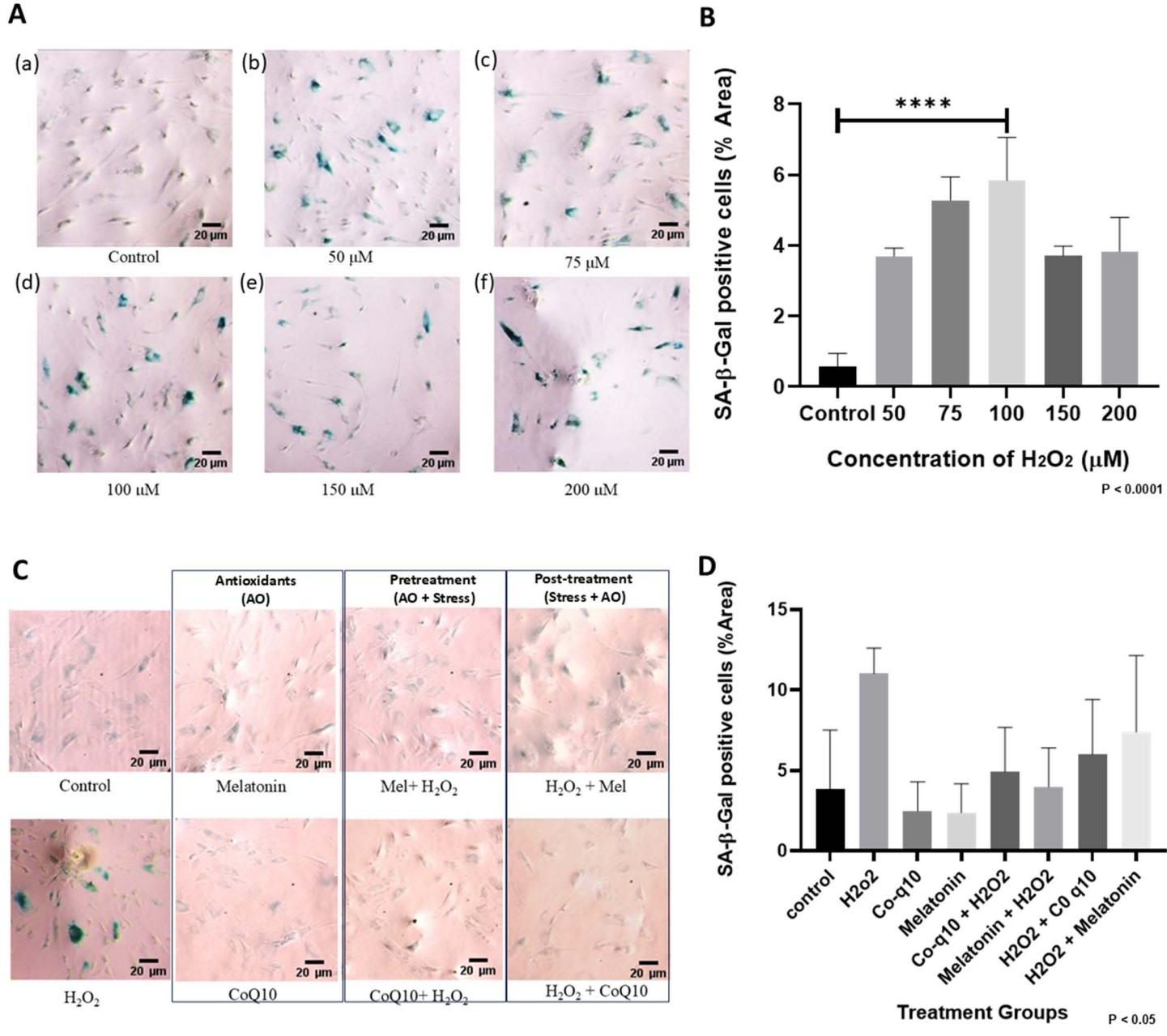

**Fig 2. Antioxidant treatment reduces cellular senescence.** SA-$\beta$-Galactosidase activity Assay in HADMSCs: **(A)** $H_2O_2$ (1 hour exposure). **(B)** Quantification of SA-$\beta$-gal (P < 0.0001; n = 3). **(C)** Comparison of pretreated and post-treated samples after 72 hours. **(D)** Quantification of SA-$\beta$-gal (P < 0.05; n = 3). Image quantification was done by ImageJ software. The results were subjected to one-way ANOVA. Post hoc analysis by Tukey's test (**** = P < 0.0001). Image at 10x- scale bar 20μm.

rather than reduced senescence (Fig 2B). Both antioxidant regimens mitigated this stress-induced senescence phenotype, as pretreatment and post-treatment with CoQ10 and melatonin have shown to reduce the proportion of SA-$\beta$-Gal-positive cells compared to the group that was given the stress alone (Fig 2C). Analysis of staining intensity showed a greater reduction in senescent area with pretreatment than with the post-treatment, with relative decrease of about

17.8% for CoQ10 and 46% for melatonin when comparing pre- vs. post-treatment conditions (Fig 2D). When stressed cells were compared directly with their respective pre-treatment groups, SA-$\beta$-Gal positive area was reduced by approximately 55.4% and 64% for CoQ10 and melatonin, respectively. (Fig 2D) (P<0.05) (One-way ANOVA). The observations from SA-$\beta$-Galactosidase staining indicates that antioxidant pretreatment confers a superior protective effect against $H_2O_2$-induced senescence.

### 3.3 CoQ10 & melatonin pretreatment of HADMSCs enhanced the intrinsic antioxidant defence mechanism and reduced ROS accumulation when stress is induced

The DCFH-DA imaging revealed that the antioxidant pretreatment has significantly attenuated ROS accumulation, even after subsequent stress exposure (Fig 3A). Quantified fluorescence intensity showed marked reduction of about 65.5% and 34.8% in cells pretreated with CoQ10 and melatonin respectively, compared to stressed cell group (P<0.0001) (Fig 3B). With regard to antioxidant scavenging capacity, both CoQ10 and melatonin pretreatment enhanced free radical scavenging potential relative to stressed cells. DPPH assay of spent media showed that the antioxidant supplementation increased extracellular radical-scavenging activity, whereas subsequent stress exposure has reduced this effect. Media from CoQ10 and melatonin-treated HADMSCs exhibited higher DPPH inhibition than serum-free medium (SFM), untreated control and stressed (STR) groups, indicating enhanced release or it could be preservation of antioxidant capacity under basal conditions. In contrast, media from cells that were antioxidant-treated prior to stress (Q10+STR) & (Mel+STR) showed intermediate DPPH inhibition, higher than the stressed group but lower than the antioxidant only group, suggesting that pretreatment partially maintains, but does not fully restore extracellular antioxidant activity in the presence of oxidative stress. DPPH inhibition increased by approximately 31% in antioxidant-pretreated groups for both compounds (Fig 3C). Similarly, consistent with strengthened cellular defence against oxidative damage, ABTS Radical Scavenging Activity (RSA%) rose by 40% for CoQ10 and 45% for melatonin pretreated cells compared to that of stressed cells (P=0.0376 and P=0.0128, respectively) (Fig 3D). These findings demonstrate that antioxidant pretreatment not only prevents ROS buildup but also restores redox balance through enhanced radical scavenging capacity.

### 3.4 CoQ10 & melatonin supplementation increased endogenous GSH and GST and supported regulation of enzymes that protect cells from oxidative stress

Antioxidant pretreatment preserved endogenous enzyme activity and reduced oxidative damage markers in stressed HADMSCs. Specifically, with regard to oxidative stress enzyme levels, SOD activity increased significantly under oxidative stress conditions but was substantially rescued by antioxidant pretreatment (Fig 4A). Normalised SOD levels were lowest in untreated controls, which rose markedly in the stressed group (STR) and were partially mitigated in cells pretreated with antioxidants CoQ10 and melatonin (P<0.0001). These findings demonstrate that antioxidant pretreatment prevents stress-induced SOD overexpression, consistent with broader antioxidant homeostasis restoration. CAT activity remained relatively stable across treatment groups, with significant increase in the stressed group (Fig 4B). Antioxidant-supplemented cells exhibited higher activity compared to the untreated control, whereas pretreated stressed group showed elevated levels than the untreated control, but lesser than the stressed group, indicating that antioxidant supplementation sustains or slightly enhances CAT activity under oxidative challenge. Oxidative stress induced marked SOD over-expression (Fig 4A), reflecting superoxide burst, while CAT activity showed compensatory elevation (Fig 4B), indicating that antioxidant pretreatment significantly attenuated SOD upregulation and maintained CAT at intermediate levels, showing coordinated ROS scavenging that reduces enzyme burden while preserving detoxification capacity. GSH levels were fairly maintained compared to the stressed group (Fig 4C), and GST levels increased in antioxidant-treated cells compared to stressed, where GST showed pronounced elevation in CoQ10-supplemented group (Fig 4D). This co-ordinated upregulation indicates enhanced detoxification capacity. These enzymes also contribute to lower $H_2O_2$ accumulation, as evidenced by reduced microlevel peroxide in antioxidant-treated cells (Fig 4E), paralleling a significant decrease in lipid peroxidation (Fig 4F). Together, these data

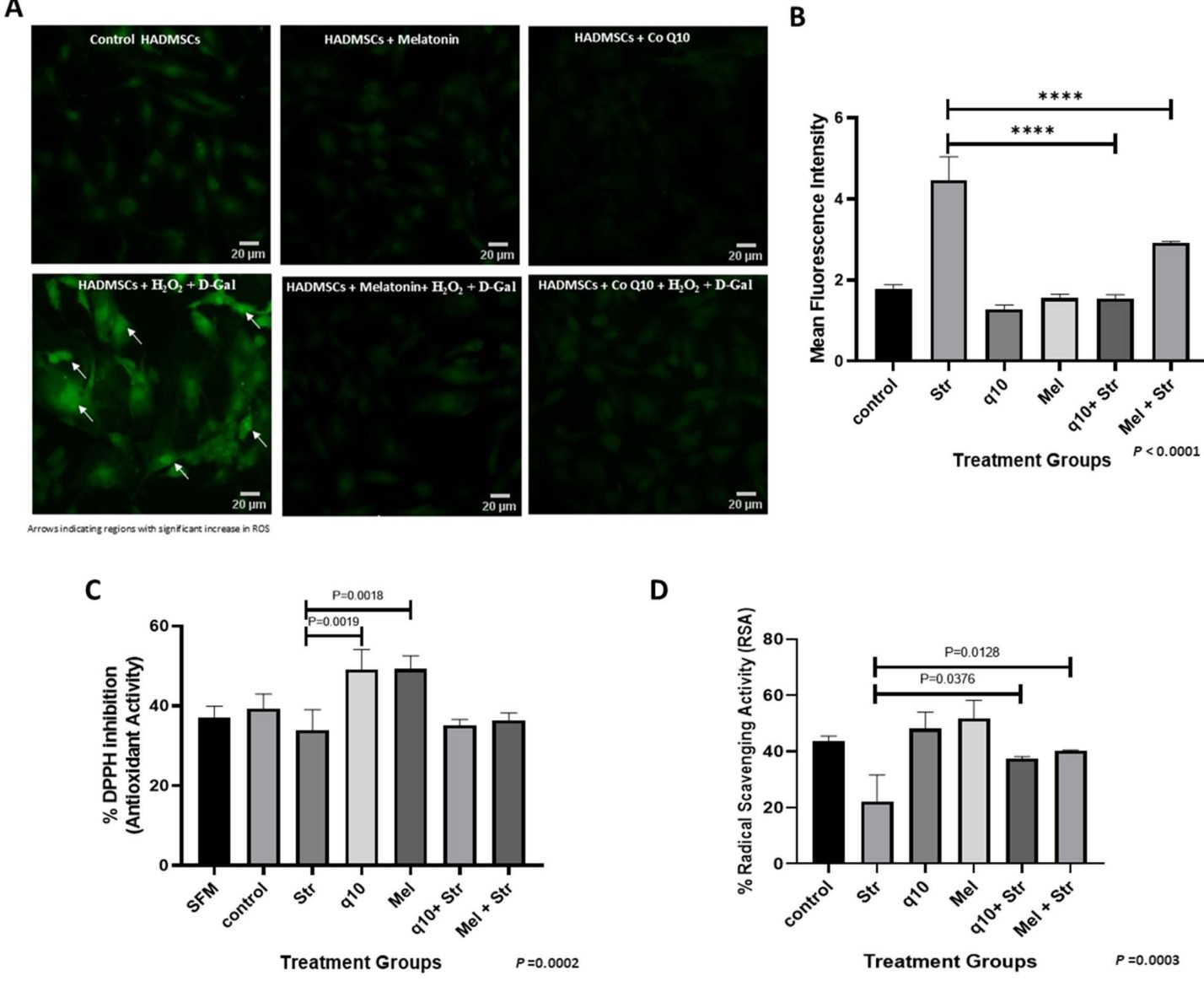

**Fig 3. Effect of Antioxidant interventions in intracellular ROS generation and cellular antioxidant capacity. (A)** Epifluorescence DCFH-DA staining image with 10X magnification. **(B)** Mean fluorescent intensity graph of DCFDA by ImageJ (P < 0.0001; n = 3). Antioxidant capacity of the cellular system: **(C)** DPPH (cell free system) and with **(D)** ABTS (cell lysate). Data are mean ± SEM (n = 3), P = 0.0002 for DPPH inhibition % and P = 0.0003 for ABTS Radical Scavenging Activity (RSA%). Statistical significance was performed using one-way ANOVA. Post hoc analysis by Tukey's test (**** P < 0.0001). Image at 10x- scale bar 20μm.

demonstrate that pretreatment restores redox homeostasis through coordinated upregulation of multiple antioxidant enzyme and substrates, mitigating ROS build-up and membrane damage under oxidative and glycative stress.

### 3.5 CoQ10 & melatonin pretreatment preserves mitochondrial membrane potential (MMP)(ΔΨm) and reduce stress-induced lysosomal accumulation even after exposure to stress induced by $H_2O_2$ and D-galactose

The protective effects of antioxidants on mitochondrial homeostasis were assessed with JC-1 and acridine orange (AO) acidic vacuole staining. The JC-1 imaging of mitochondria in the samples after treatment revealed that treatment with

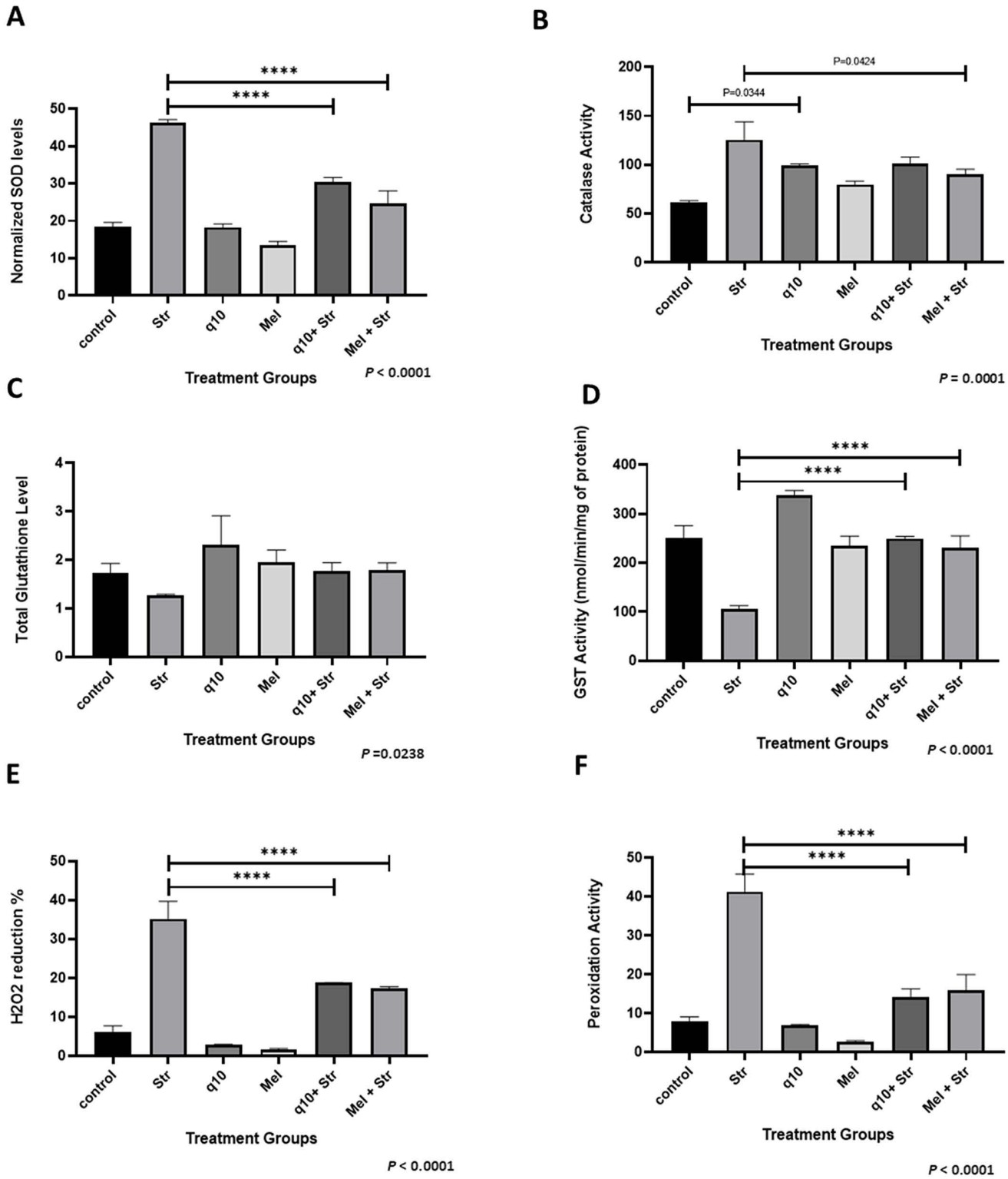

**Fig 4. Regulation of oxidative stress enzymes.** ROS assays for the treatment groups (24 hours post-treatment): **(A)** SOD assay, **(B)** CAT assay, **(C)** GSH assay, **(D)** GST assay, **(E)** Micro $H_2O_2$ assay, **(F)** Lipid peroxidation assay. Data are mean ± SEM ($n = 3$), and the results were subjected to one-way ANOVA ($P < 0.0001$). Post hoc analysis by Tukey's test (**** = $P < 0.0001$).

D-gal and $H_2O_2$ has significantly weakened the MMP (green monomer accumulation), as indicated by white arrows (Fig 5A). Pre-treatment with antioxidants significantly helped to mitigate mitochondrial membrane damage and preserved the MMP by 61.7% and 58.4% for CoQ10 and melatonin, respectively, with higher red J-aggregate fluorescence compared to stressed cells ($P < 0.001$) (Fig 5B). Whereas, antioxidant pretreatment is observed to have helped in the maintenance of MMP similar to control cells, even after exposure to stress. Cells stained with AO showed autophagic vesicles and acidic vacuoles formed within the cells as red fluorescence. The percentage of red fluorescence in stress-induced group was significantly higher and was observed to have distorted morphology (Fig 5C). Control cells showed moderate green cytoplasmic fluorescence with few distinct red/orange puncta, indicating basal lysosomal activity. Antioxidant-supplemented cells show reduced red fluorescence compared to that of the control, indicating the effect of antioxidants in lysosomal homeostasis. In antioxidant-pretreated cells, decreased red fluorescence was observed compared to the stress alone (STR), which could indicate mitigation of oxidative stress-induced acidic vacuole formation and provision of partial protection against lysosomal stress. ImageJ software analysis of AO staining showed that, the red/green fluorescence ratio has a 2.23-fold and 3.3-fold decrease in healthy cells supplemented with antioxidants compared to that of the control for CoQ10 and melatonin, respectively ($P < 0.01$ & $P < 0.001$) (Fig 5D). There was also a decrease in pretreated cells about 3.3-fold (CoQ10 + STR) & 3.0-fold (Mel + STR) when compared to the stress group, indicating lesser acidic vacuole formation accumulation in pretreated cells ($P < 0.0001$).

### 3.6 CoQ10 & melatonin pretreatment of HADMSCs modulated the expression of mitophagy receptors and LC3B in stressed HADMSCs

Western blot analysis of protein lysates collected from HADMSCs subjected to different treatments revealed distinct patterns of mitophagy receptor expression (Fig 6A–6G). Parkin levels increased in antioxidant-supplemented cells whereas the stressed group showed no detectable accumulation (Fig 6B). Similarly, stress-induced upregulation of optineurin was attenuated by antioxidant treatment (Fig 6C). $H_2O_2$ and D-galactose stress significantly reduced NDP52 levels, while antioxidant intervention has effectively preserved its expression (Fig 6D). BNIP3 protein expression decreased markedly in stressed cells but was substantially restored in CoQ10-pretreated stressed HADMSCs (Fig 6E). In contrast, BNIP3L/Nix levels peaked in control & CoQ10 supplemented cells, and were downregulated under stress conditions (Fig 6F). LC3B protein levels increased significantly across all treatment groups relative to untreated control (Fig 6G). Visual quantification of ubiquitin-dependent and ubiquitin-independent mitophagy protein expression in HADMSCs under antioxidant intervention and oxidative stress (Fig 7).

### 4. Discussion

The antioxidant capacity of endogenous molecules such as CoQ10 & melatonin- both naturally produced by cells but depleted with age, has been extensively studied for mitigating oxidative damage, where supplementation has been shown to preserve MMP, enhance endogenous antioxidant enzyme activity, promote mitophagy, increase mitochondrial mass, modulate permeability transition pore opening, suppress ROS generation and improve overall mitochondrial bioenergetics [34,35]. However, the anti-aging potential of antioxidants warrants re-evaluation, as high doses may transiently inhibit proliferation in quiescent cells without genotoxicity, whereas in proliferating cells, they can induce irreversible cell cycle arrest, DNA damage, and premature senescence [16]. Given the critical influence of intervention timing on cellular signalling, we employed both —pretreatment (antioxidant administration prior to stress) and post-treatment (antioxidant administration following stress) strategies to systematically evaluate their comparative efficacy in mitigating senescence in HADMSCs.

The MTT assay demonstrated improved proliferative capacity of HADMSCs by intervention with concentration as low as $10^{-7}$M. Also, cells exhibited superior proliferative capacity in pretreated cells compared to post-treated counterparts. However, viability declined in a dose-dependent manner at concentrations exceeding $10^{-6}$M. $H_2O_2$, a widely used inducer of oxidative stress in cell cultures, functions as a signaling molecule at moderate levels [5], but induces cellular damage at

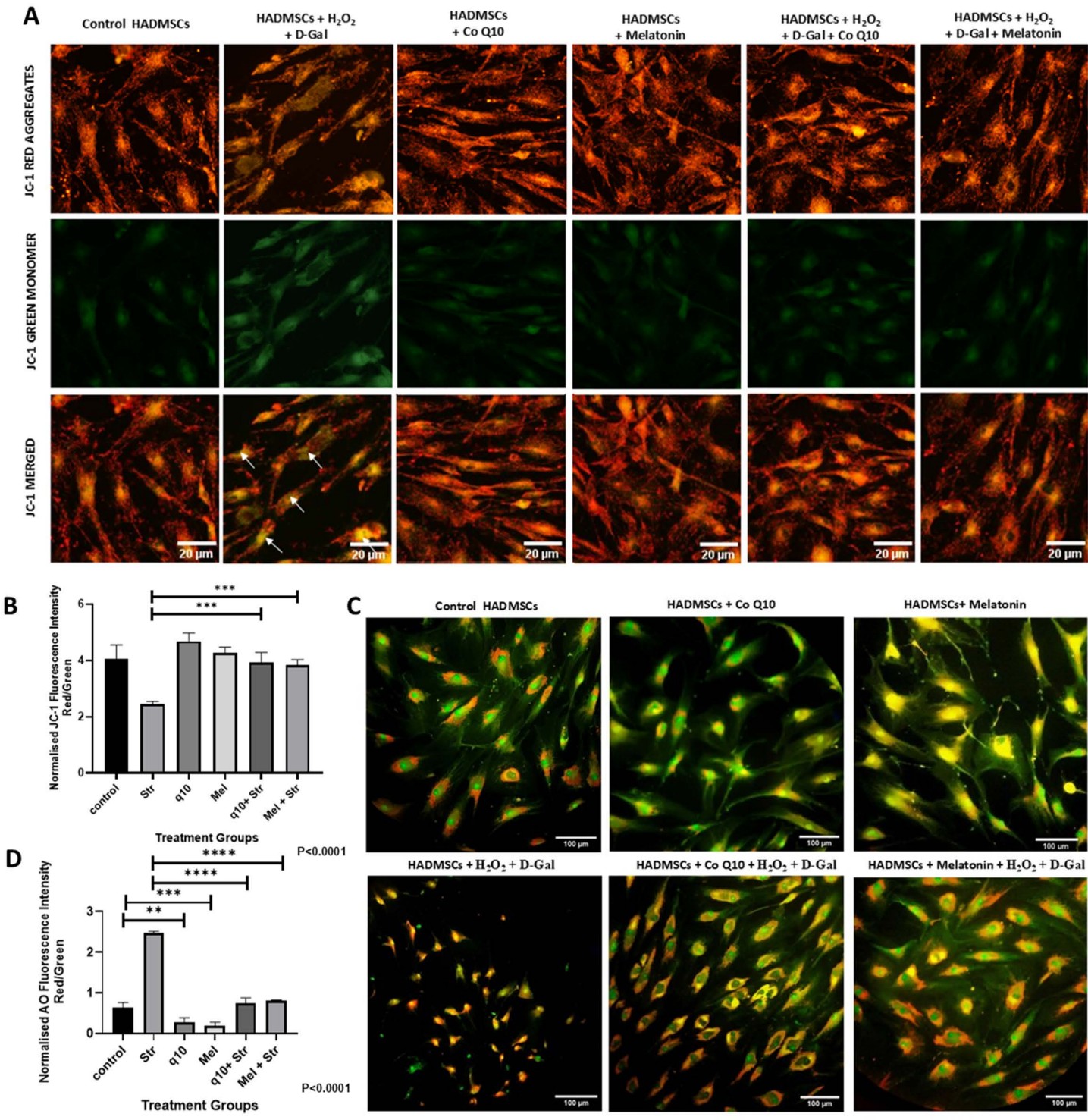

**Fig 5. Preservation of mitochondrial membrane potential (MMP) by antioxidants and alleviation of lysosomal accumulation induced by oxidative and glycative stress.** Fluorescence Imaging and Quantitation: Detection of MMP (ΔΨm) with **(A)** JC-1 imaging showing MMP loss(green) in stressed cells (white arrows). Image at 40x- scale bar 20μm, **(B)** Quantification of the JC-1, **(C)** Acridine Orange staining showing red acidic vacu-oles (AO). Image at 20x- scale bar 100μm. **(D)** Quantification of AO by red/green fluorescence ratio. Images were quantified using ImageJ software. Data are mean±SEM (n=3), (P<0.0001). Statistical significance was calculated using one-way ANOVA. Post hoc analysis by Tukey's test (**P<0.01; ***P<0.001; ****P<0.0001).

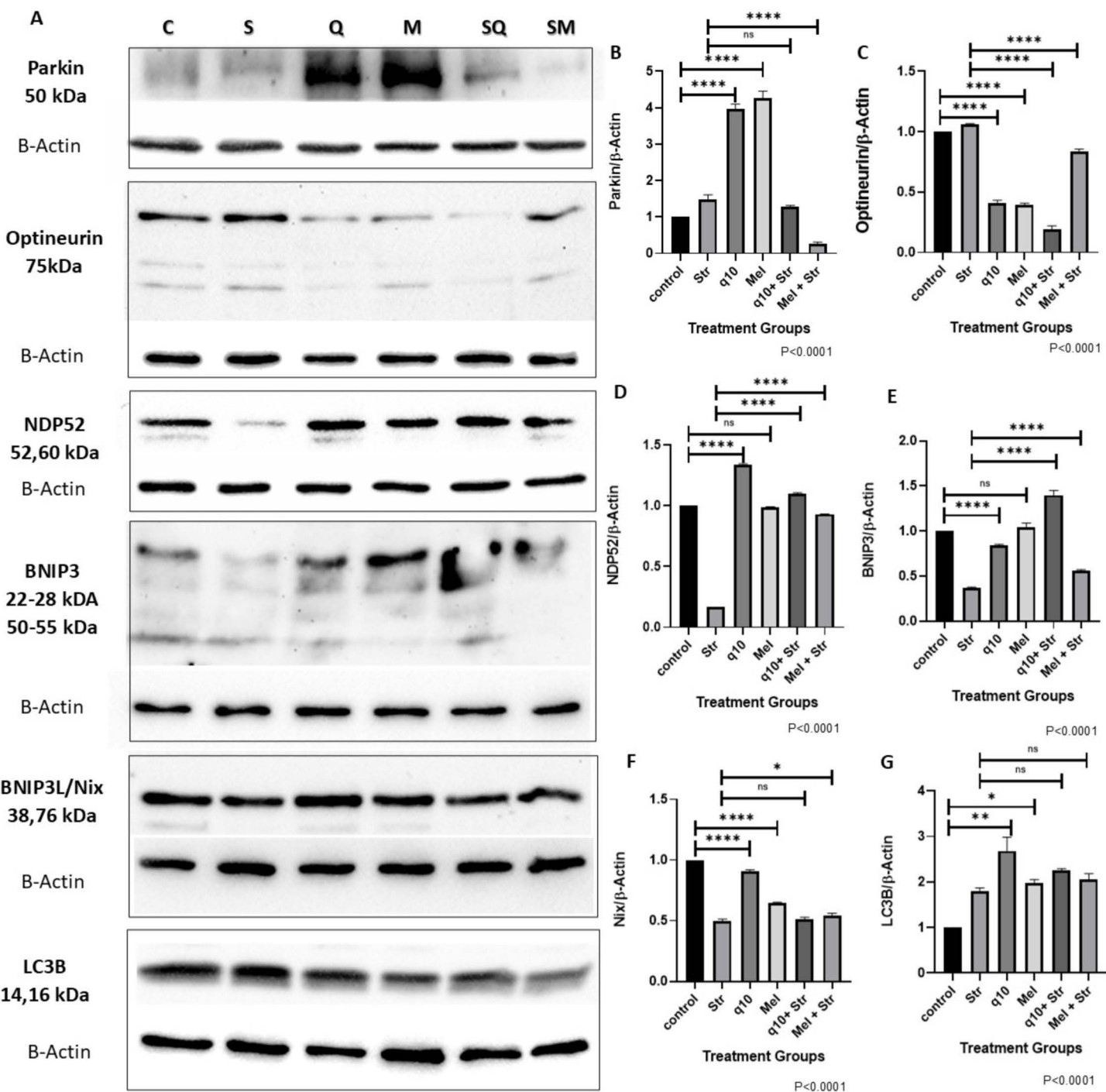

**Fig 6. Modulation of mitophagy-related protein expression.** Western Blot expression of mitophagy proteins: **(A)** Chemiluminescent imaging of antibodies by Bio-Rad ChemiDoc. Full blot images available in Supplementary File: S1 Fig – Complete and uncropped Western blots. Fig(B–G) Densitometry of different treatment groups performed using ImageJ software (normalized with B-actin expression) **(B)** Parkin (P < 0.0001), **(C)** Optineurin (P < 0.0001), **(D)** NDP52 (P < 0.0001), **(E)** BNIP3 (P < 0.0001), **(F)** BNIP3L/Nix (P < 0.0001), **(G)** LC3B (P = 0.0026). Data are mean ± SEM (n = 3), normalized to control (100%). Statistical significance was analyzed using one-way ANOVA (n = 3). Post hoc analysis was performed using Tukey's test for comparison across different treatments. (*P < 0.05; **P < 0.005; ****P < 0.0001).

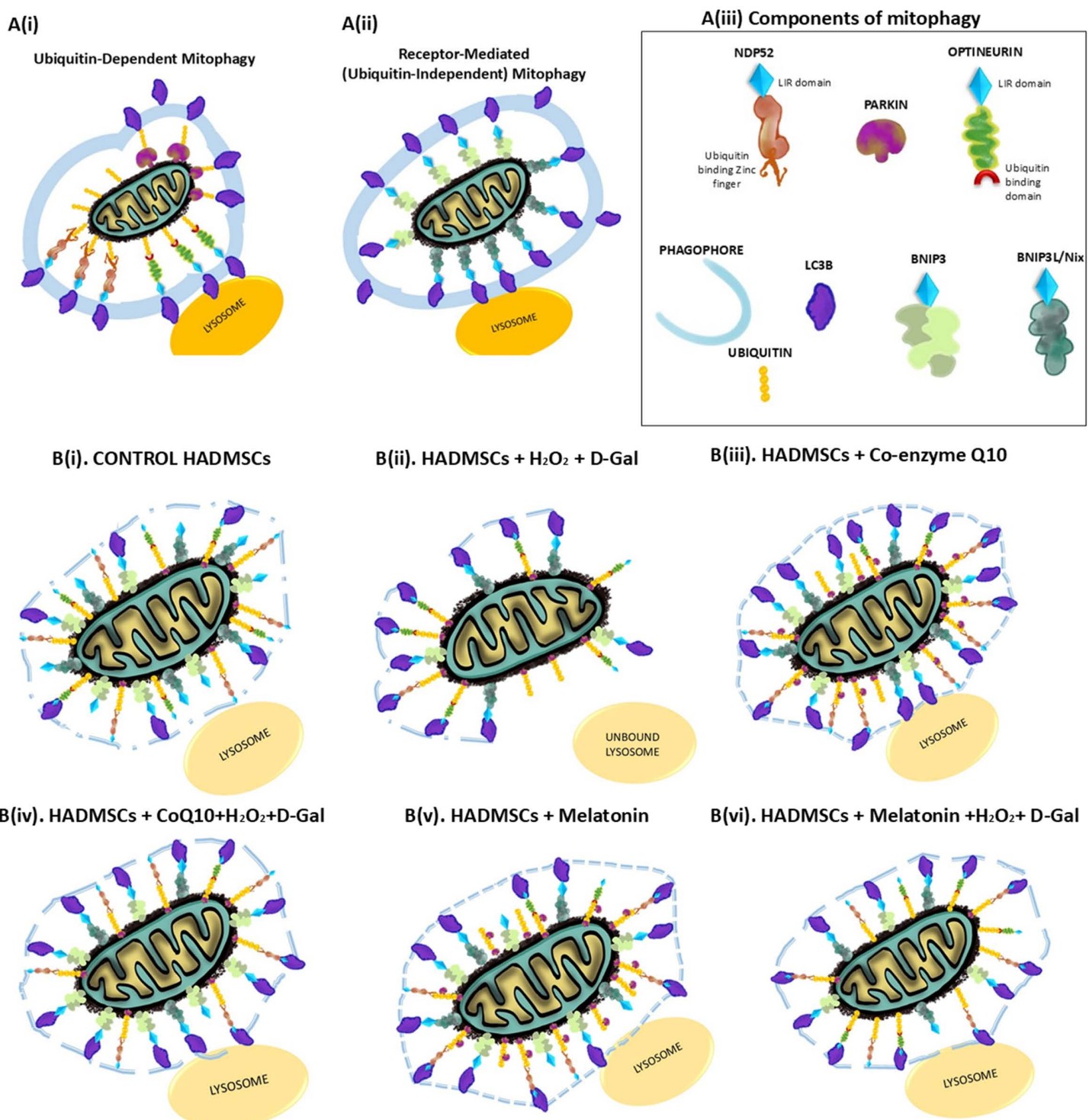

**Fig 7. Protein Portraits- Representation of Ubiquitin dependent and Ubiquitin independent (receptor mediated) Mitophagy.** Visual Quantification of Mitophagy Protein Expression Patterns in HADMSCs by Antioxidant Intervention and Oxidative Stress Induction. **(A)** Pictorial illustration of mitophagy proteins: A(i) Ubiquitin-dependent proteins: NDP52, Parkin, and Optineurin with LC3B (autophagy machinery), A(ii) Ubiquitin-independent or receptor-mediated proteins: BNIP3 and BNIP3L/Nix with LC3B, A(iii) Mitophagy protein components. The elements in the image were sketched using Adobe Fresco and were arranged with Microsoft PowerPoint. **(B)** (i)-(vi) The expression levels of proteins through Western blot were quantitatively measured and represented by a unique pictographic element for each protein observed in this study.

higher concentrations [7]. Given the susceptibility of *in-vitro*-expanded MSCs to ROS-mediated senescence, we optimized stress conditions using MTT assay and SA-β-gal staining across $H_2O_2$ concentrations and exposure durations (2 hours, 1 hour, and 30 minutes). A 100 µM $H_2O_2$ exposure for 1 hour proved optimal for subsequent studies, balancing sublethal stress with senescence induction. D-Galactose (100mM), an established accelerator of aging phenotypes, [36] was combined with $H_2O_2$ to model compounded oxidative/glycative damage in HADMSCs.

SA-β-Gal staining revealed a significant reduction in SA-β-Gal-positive cells following antioxidant pre-treatment, demonstrating effective suppression of senescence phenotype. DCFH-DA fluorescence imaging confirmed elevated ROS levels in stressed cells, which were markedly attenuated by low-dose antioxidant pretreatment, indicating that pre-emptive intervention effectively limits ROS accumulation. Given the challenge of poor survival and engraftment rates of mesenchymal stem cells at injured sites—primarily attributable to inflammation and oxidative stress [5] — antioxidant preconditioning of HADMSCs emerge as a promising strategy to enhance cellular resilience and therapeutic potential.

DPPH, which is a stable free radical, was used here to measure the secreted antioxidants present in the media, because as a result of cell metabolism, certain antioxidants can be secreted out of cells under specific conditions like starvation or in response to ROS [37]. DPPH assay of spent media revealed no significant differences in radical scavenging activity among control, stressed cells, and antioxidant-pretreated cells, with only modest increase in antioxidant-supplemented groups. These finding suggest that secreted antioxidants or residual supplemented compounds contribute minimally to extracellular antioxidant capacity under the tested conditions. This further needs to be strategically analyzed to determine whether antioxidant supplementation enhance paracrine protection through secreted antioxidants. If the hypothesis of antioxidant secretion holds good, then such secretion could be harnessed to protect the cellular microenvironment, given that superoxide radicals exhibit short half-lives and limited diffusion distances causing localized damage at the site of generation, whereas $H_2O_2$ diffuses further to serve as signaling molecules, potentially propagating oxidative damage and DNA damage to neighboring cells [38].

In contrast, ABTS assay performed on cell lysate to quantify the total antioxidant capacity (enzymatic and non-enzymatic antioxidant) of cells, demonstrated significantly elevated levels of RSA% in the antioxidant-pretreated cells compared to the stressed group, indicating enhanced intracellular antioxidant potential.

Low levels of ROS are essential for self-renewal of stem cells [9]. For clinical translation, *in-vitro* expansion of stem cells becomes indispensable to obtain therapeutic cell numbers, yet prolonged culture induces ROS-mediated senescence leading to loss of stemness [8], ultimately compromising post-transplantation efficacy of MSCs[6]. Exogenous antioxidants not only directly scavenge ROS, but also augment cellular antioxidant machinery [12], including SOD, CAT, GSH, and GST, and directly scavenge micro-$H_2O_2$ released, thereby mitigating lipid peroxidation and preserving stem cell functionality during expansion.

SOD levels peaked in stressed HADMSCs, reflecting a compensatory response to superoxide accumulation, while antioxidant pretreatment significantly attenuated upregulation compared to that of stress ($P < 0.0001$). Even though SOD's primary role is to neutralize ROS, higher SOD levels can disrupt normal cellular processes, leading to consequences like potential cell damage and altered cell behavior [39]. An increase in SOD also leads to the production of higher micro-$H_2O_2$ levels due to the dismutation of superoxide, which needs to be neutralized by the catalase enzyme. Catalase activity showed modest elevation across treatment groups, with a statistically significant increase in CoQ10-supplemented cells compared to untreated control ($P = 0.0344$), and in melatonin-pretreated cells prior to stress compared to stress alone ($P = 0.042$). The stressed group had the highest value, which can be correlated with high SOD levels. Antioxidant-supplemented and pretreated cells showed elevated CAT levels compared to the controls, where these patterns indicate compound-specific modulation under oxidative/glycative challenge. The mechanism behind these dynamics is not clearly understood and needs further investigation.

Glutathione (GSH) serves as a critical antioxidant essential for neutralizing ROS within cells, where elevated levels of GSH have been shown to safeguard against detrimental DNA damage as they act as redox buffers [9]. Melatonin and GSH supplementation have been reported to effectively prevent dysfunction and preserve cell function in long-term

                                       

passaged HADMSCs[8]. Melatonin supplementation has been shown to increase the levels of cellular glutathione (GSH) and enhanced GSH synthesis and glutathione cycle-related enzymes, ultimately boost the body's antioxidant defenses [13]. Some studies suggest that CoQ10 supplementation can increase serum glutathione peroxidase (GSH-Px) levels, an enzyme involved in antioxidant defense, whereas others demonstrate no significant impact on GSH levels or glutathione peroxidase activity [40]. In our observations, the GSH levels displayed modest variation across groups, with very little significant deviation in comparison with control versus antioxidant-supplemented and stressed versus antioxidant-pretreated cells; significance was observed only between the stressed and CoQ10-supplemented groups. These findings indicate selective modulation of GSH pools by supplementation under oxidative/glycative challenge, which needs to be further investigated.

In this context, low levels of GST in stressed cells could be due to overwhelmed SOD and CAT levels, which could also be correlated with lesser GSH levels in the stressed cells, where GSH acts as a cofactor for GST [41]. We found a substantial increase in GST activity in antioxidant-pretreated cells compared to stressed cells ($P < 0.0001$), despite modest GSH level variation, consistent with enhanced phase II detoxification capacity. The antioxidant pretreatment was able to maintain the endogenous GSH levels, which helped maintain GST levels despite induction of stress. Micro-levels of $H_2O_2$ within the cells revealed prominent reduction in their levels in the antioxidant-supplemented and pretreated group in comparison to control and stressed groups respectively ($P < 0.0001$). Their levels could be correlated with the lipid peroxidation, as antioxidant intervention significantly reduced lipid peroxidation. Reduction in GST levels in the stressed group and high $H_2O_2$ could be a major factor leading to increased lipid peroxidation in HADMSCs.

Here, normalized JC-1 red/green fluorescence ratio, indicative of mitochondrial membrane potential (MMP), decreased markedly in stressed cells relative to the controls. Pretreatment with antioxidants prior to stress revealed that antioxidant-pretreatment significantly protected the mitochondrial membrane in HADMSCs after $H_2O_2$ and D-gal exposure compared to that of the stressed group and was maintained similar to that of the untreated control, restoring the ratio towards control levels. Antioxidant-supplemented cells without stress maintained only a slightly better ratio, which could be due to the cells' innate threshold of maintenance of MMP, tightly regulated by a complex interplay of factors, including the electron transport chain (ETC), ATP synthase, and various membrane transporters [42].

Autophagic vacuoles in stem cells can trigger apoptosis, and with excessive autophagy potentially can cause cell death, while also promoting survival by removing damaged components [32]. Antioxidant intervention has significantly minimized acidic vacuole accumulations in HADMSCs ($P < 0.0001$). Control cells have a basal level of acidic vacuoles too. Pretreated cells showed healthy levels of acidic vacuole formation as a result of antioxidant intervention. Melatonin enhances ATP production, reduces ROS, and suppresses senescence-derived mitochondrial dysfunction through mitophagy [34]. Mitophagy not only removes damaged mitochondria; in certain cases of stem cells, active mitochondria are removed to maintain low metabolic activity in stem cells, thus enhancing the regenerative capacity of aging stem cells [43]. Mitophagy tends to reduce over time, leading to a decline in autophagic efficiency, which underscores the importance of maintaining mitophagy to ensure optimal stem cell function and overall tissue homeostasis as organisms age [10]. Balance in autophagy and mitophagy is critical for stem cell health and prevention of age-related diseases, as it directly influences cellular resilience and functionality [10]. To combat impaired mitophagy, induction of mitophagy could be a therapeutic intervention to impede aging by targeting the clearance mechanism of dysfunctional mitochondria [43]

Cells require autophagy receptors like NDP52 and optineurin proteins to identify and selectively degrade ubiquitinated cargo. Parkin then binds them to the LC3 on the phagophore membrane, whereas BNIP3 and BNIP3L/Nix are membrane proteins that directly interact with LC3 [44]. Melatonin affects Parkin expression in stem cells, promoting mitophagy [45], and CoQ10 enhances mitophagy flux, improving cell pathophysiology and parkin-mediated mitophagy [46]. Here, parkin levels significantly increased in antioxidant-supplemented cells in comparison to the control group, indicating parkin-mediated mitophagy ($P < 0.0001$). Here, Parkin levels of CoQ10-pretreated cells did not show much significance when compared to that of the stressed cells, whereas melatonin-pretreated cells showed significant reduction in Parkin

expression compared to that of the stress group (P<0.0001). $H_2O_2$ is known to cause dimerization of parkin at a molecular level, which causes aggregation leading to the formation of high molecular weight complexes, also leading to polyubiquitination [47]. Studies have shown that D-galactose administration decreased PINK1 and Parkin expression [48]

Optineurin (OPTN) is a protein known to be involved in autophagy, vesicular transport, and ER stress and delays senescence in bone marrow MSCs [49]. Optineurin's recruitment to mitochondria is crucial for linking parkin activity to autophagosome formation during mitophagy, where it stabilizes damaged mitochondria and recruits LC3, resulting in autophagosome formation around mitochondria; disruption of which leads to mitochondrial damage and accumulation [50]. In our studies, there is no significant difference found in the expression levels of optineurin in control and stressed cells, whereas antioxidant supplementation has reduced the expression significantly (P<0.0001). In antioxidant-pretreated cells, optineurin expression in CoQ10-pretreated stressed cells significantly dropped compared to that of melatonin-pretreated stressed cells. Overall, optineurin protein levels were found to be reduced with antioxidant intervention shows a shift from optineurin-dependent mitophagy to alternate pathways which explains preserved mitochondrial health despite lower optineurin levels.

NDP52 is a crucial protein in PINK1/Parkin-mediated mitophagy, recognizing signals from ubiquitinated surface proteins and initiating autophagy while functioning as a redox sensor [51]. We observed that NDP52 expression was lowered in the stress-induced group, and antioxidant intervention significantly helped in the maintenance of NDP52 levels without being affected by $H_2O_2$ and D-Gal stress. CoQ10 supplementation showed a significant increase in expression. In our studies, optineurin expression increased, whereas NDP52 reduced in the stressed group, indicating the effect of D-gal and $H_2O_2$ on mitophagy cargo proteins. Optineurin was significantly reduced in antioxidant-treated groups in comparison to the control, whereas NDP52 expression increased.

Studies show that melatonin affects stem cells by affecting autophagy gene expression, particularly BNIP3 and Nix, which can lead to increased BNIP3 expression, promoting cell health [52]. CoQ10 may enhance mitochondrial health and potentially benefit from BNIP3-mediated mitophagy in conjunction [15]. BNIP3 protein expression was significantly reduced in the stress group (STR), whereas it was observed to be highly upregulated in the CoQ10-pretreated group, indicating activation of BNIP3 pathway in CoQ10 pretreatment group. When CoQ10-pretreated cells were exposed to stress, BNIP3 expression increased, indicating mitophagy mediated through BNIP3. BNIP3L/Nix expression was observed to be the highest in control and CoQ10-supplemented cells. It was found to be downregulated in stress-induced groups, indicating that $H_2O_2$ and D-galactose affected BNIP3L/Nix expression irrespective of antioxidant pretreatment. The antioxidant-supplemented group and pretreated group showed increased expression, indicating activation of BNIP3L/Nix mitophagy of damaged mitochondria. LC3B expression increased in all treated groups in comparison to the control, where the context of increased expression differs in each group, indicating that they are facilitating mitophagy; whereas in the stressed group there was no accumulation of NDP52, indicating overwhelming damage causing impaired NDP52 and sub-optimal Parkin expression.

The dynamics of mitophagy proteins were found to be altered, indicating the multifaceted approach of HADMSCs and antioxidants in their survival, depending on the signaling proteins within the system. Mitophagy and autophagy are critical in senescence, yet their interplay has a significant function in the clearance of damaged mitochondria along with the other damaged cellular organelles. LC3B accumulation is not specific to mitophagy alone, it also indicates autophagy, whereas BNIP3 and NIX are mitochondrial membrane proteins, and NDP52 and optineurin are multifunctional cargo proteins that aid mitophagy.

Mitochondria respond to energy deficiency through retrograde signaling, triggering metabolic adaptation, compensatory mitochondrial biogenesis, antioxidant enzyme upregulation, and protein acetylation, and also mitophagy triggers metabolic shifts, while antioxidants control ROS levels in stem cells, contributing to metabolic re-programming [53,54]. To further improve the understanding of antioxidant intervention in HADMSCs, biomarkers related to inflammation and mitochondrial dynamics could be assessed. There are multiple parameters that affect the quality of HADMSCs, where special attention

is required with regard to the passage number of the cells and dose- and time-dependent analysis to validate the intricate protein dynamics within the cells. Understanding the process through all these dimensions will give us a collective perspective to successfully reverse aging and will boost up regenerative therapies.

## 5. Conclusion

This study highlights the critical interplay among oxidative stress, antioxidant intervention, mitophagy, and senescence in HADMSCs. The protein initiating the specific pathway of mitophagy has been observed to vary across the treatment groups. A plausible reason could be the difference in the mechanism of action of CoQ10 and melatonin, or the redox state of cells might alter the trigger to different proteins. The connection between the trigger and the regulator is found to be quite intricate and yet to be clearly understood. Our findings suggest that melatonin pretreatment prior to stress enhanced mitophagy through optineurin and NDP52 pathway, whereas CoQ10 pretreatment prior to stress has enhanced mitophagy through NDP52 and BNIP3. Supplementation without stress induction revealed that melatonin downregulated optineurin and BNIP3L/nix, whereas it upregulated parkin and BNIP3 expression. In CoQ10 supplementation without stress induction, it was observed that parkin and NDP52 were upregulated, and optineurin was downregulated. Upregulated LC3B expression with antioxidant interventions, indicates enhanced autophagosome formation capacity, concurrent with mitophagy receptor modulation. This indicates broad autophagy priming, that enable selective mitophagy through available receptors, which explains the preserved mitochondrial health despite stress. Antioxidant intervention improved the functional state and proliferation of HADMSCs by the maintenance of mitochondrial integrity, regulation of oxidative stress enzymes and mitophagy proteins, which ultimately mitigates senescence. These results underline the potential of mitochondrial quality control to delay senescence of stem cells through antioxidant intervention. Further *in-vivo* validation and mechanistic dissection could pave the way for novel perspectives on CoQ10 and melatonin intervention for anti-aging and regenerative medicine.

## Supporting information

**S1 Fig. Full-length Western Blots of mitophagy-related proteins in HADMSCs. Representative uncropped and unedited Western blot images for all proteins analyzed in the study (Parkin, Optineurin, NDP52, BNIP3, BNIP3L/Nix, and LC3B) under different treatment conditions with antioxidant interventions in oxidative stress conditions. Blots were probed with the indicated primary antibodies and developed using enhanced chemiluminescence; molecular-weight markers which are shown on the left in kilodaltons (kDa).**
(PDF)

**S2 Fig. Rescuing HADMSC Senescence through Antioxidants via Enhanced Mitophagy.**
(TIF)

## Author contributions

**Conceptualization:** Anuradha Dhanasekaran.

**Data curation:** Aleena Vikraman.

**Formal analysis:** Aleena Vikraman.

**Funding acquisition:** Aleena Vikraman.

**Investigation:** Aleena Vikraman, Logeswari Ravi.

**Methodology:** Aleena Vikraman, Logeswari Ravi, Anuradha Dhanasekaran.

**Project administration:** Anuradha Dhanasekaran.

**Resources:** Anuradha Dhanasekaran.

**Software:** Aleena Vikraman.

**Supervision:** Anuradha Dhanasekaran.

**Validation:** Anuradha Dhanasekaran.

**Visualization:** Aleena Vikraman, Logeswari Ravi.

**Writing – original draft:** Aleena Vikraman.

**Writing – review & editing:** Naveena Kandasamy, Anuradha Dhanasekaran.

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
