## [Decision Letter · Decision Letter 0]

23 Dec 2025

PONE-D-25-60616Synergistic Role of Mitophagy and Antioxidants in enhancing the fate and survival of Mesenchymal Stem Cells (MSCs) - Plausible role in Metabolic ReprogrammingPLOS One

Dear Dr. Dhanasekaran,

Thank you for submitting your manuscript to PLOS ONE. After careful consideration, we feel that it has merit but does not fully meet PLOS ONE’s publication criteria as it currently stands. Therefore, we invite you to submit a revised version of the manuscript that addresses the points raised during the review process.

Considering the reviewers comments and a careful reading of the manuscript, it has been decided that the authors need to significantly revise the manuscript and provide adequate responses to the reviewers queries.==============================

If applicable, we recommend that you deposit your laboratory protocols in protocols.io to enhance the reproducibility of your results. Protocols.io assigns your protocol its own identifier (DOI) so that it can be cited independently in the future. For instructions see: https://journals.plos.org/plosone/s/submission-guidelines#loc-laboratory-protocols. Additionally, PLOS ONE offers an option for publishing peer-reviewed Lab Protocol articles, which describe protocols hosted on protocols.io. Read more information on sharing protocols at . Additionally, PLOS ONE offers an option for publishing peer-reviewed Lab Protocol articles, which describe protocols hosted on protocols.io. Read more information on sharing protocols at https://plos.org/protocols?utm_medium=editorial-email&utm_source=authorletters&utm_campaign=protocols..

We look forward to receiving your revised manuscript.

Kind regards,

Hafiz Muhammad Umer Farooqi

Academic Editor

PLOS One

Journal Requirements:

“This work was supported by the Research Fellowship provided by University Grants Commission (UGC), Government of India, through National Eligibility Test (NET Fellowship- JRF & SRF awarded to Aleena Vikraman [(Award letter: F.No.16-6(DEC 2018)/2019(NET/CSIR); UGC Ref. no. 470/ (CSIR-UGC NET DEC 2018)”

“This work was supported by the Research Fellowship provided by University Grants Commission (UGC), Government of India, through National Eligibility Test (NET Fellowship- JRF & SRF awarded to Aleena Vikraman [(Award letter: F.No.16-6(DEC 2018)/2019(NET/CSIR); UGC Ref. no. 470/ (CSIR-UGC NET DEC 2018)”

“This work was supported by the Research Fellowship provided by University Grants Commission (UGC), Government of India, through National Eligibility Test (NET Fellowship- JRF & SRF awarded to Aleena Vikraman [(Award letter: F.No.16-6(DEC 2018)/2019(NET/CSIR); UGC Ref. no. 470/ (CSIR-UGC NET DEC 2018)”

In your cover letter, please note whether your blot/gel image data are in Supporting Information or posted at a public data repository, provide the repository URL if relevant, and provide specific details as to which raw blot/gel images, if any, are not available. Email us at plosone@plos.org if you have any questions."

Additional Editor Comments:

Considering the reviewers comments and a careful reading of the manuscript, it has been decided that the authors need to significantly revise the manuscript and provide adequate responses to the reviewers queries.

Reviewers' comments:

Reviewer's Responses to Questions

**Comments to the Author**

1. Is the manuscript technically sound, and do the data support the conclusions?

Reviewer #1: Partly

Reviewer #2: Partly

2. Has the statistical analysis been performed appropriately and rigorously? 

Reviewer #1: Yes

Reviewer #2: No

3. Have the authors made all data underlying the findings in their manuscript fully available?

Reviewer #1: Yes

Reviewer #2: Yes

4. Is the manuscript presented in an intelligible fashion and written in standard English?

Reviewer #1: No

Reviewer #2: No

5. Review Comments to the Author

Reviewer #1: This manuscript examines the effects of CoQ10 and melatonin on stress-induced dysfunction in human adipose-derived MSCs, and the authors conclude that antioxidant supplementation improves mitochondrial homeostasis, reduces cellular senescence, and enhances mitophagy, particularly through Parkin- and BNIP3-associated pathways.

While the topic is relevant, the manuscript in its current form is far from publication. The writing is difficult to follow, and the results section reads more like extended figure legends than a coherent scientific narrative. There is no logical flow in the results section and many statements are grammatically unclear, making the manuscript challenging to read.

Besides, the central conclusion—that antioxidants enhance clearance of damaged mitochondria—is not supported by the assays presented. JC-1, acridine orange staining, and changes in mitophagy-related proteins reflect alterations in mitochondrial membrane potential, lysosomal activity, and pathway signaling but do not demonstrate mitophagy flux, mitochondrial degradation, or mitochondria–lysosome colocalization. The data therefore do not establish mitochondrial clearance, and additional evidence is required to demonstrate the conclusion.

Lastly, comparisons between pre- and post-treatment conditions are also not rigorously analyzed, while the author claimed pre-treatment worked better. The interpretation of the DPPH assay is similarly incomplete, particularly regarding the lack of improvement in antioxidant-pretreated stressed cells.

Overall, the manuscript is far from publication-ready and requires extensive rewriting, clearer data interpretation, and more cautious conclusions. A full major revision followed by another round of evaluation will be necessary.

Reviewer #2: Vikraman and colleagues present a manuscript that examines the role of mitophagy in regulating senescence in human adipose-derived mesenchymal stem cells (HADMSCs) and the therapeutic potential of antioxidants melatonin and coenzyme Q10 (CoQ10). Stress-induced HADMSCs showed reduced mitochondrial membrane potential, increased ROS, and higher senescence-associated β-galactosidase activity. Antioxidant treatment reduced ROS and lipid peroxidation, restored mitophagy markers, and improved stem cell function. These findings suggest that antioxidant-induced mitophagy may reverse stem cell aging, enhance regeneration, and improve stem cell transplantation efficiency. Generally, the quality of the manuscript needs to be improved. And I have following comments:

1. The author should add page numbers and line numbers in the manuscript, so that the reviewers can clearly indicate which parts of the manuscript they are referring to.

2. In “3. Results, 3.1”, the authors stated that “(Fig.1.A) Antioxidants showed maximum growth at 10-7 M, and elicited a cytotoxic response at 10-3 M concentration. Increased proliferative potential from 10-10 M to 10-7 M range.” The “(Fig.1.A)” should be put behind the sentences. The same applies elsewhere.

3. In Figure 1, the p values were labelled in a very confusing way. The authors seemed like only to choose some of the groups to label the p values without proper explanation.

4. In figure 2A and C, there is no scale bar. The labels of p value are not fully consistent with the manuscript.

5. In figure 3A, there is no scale bar. The way of showing inconsistent with previous figure.

6. In figure 5C, the image magnification is inconsistent among groups. And there is no scale bar.

7. In figure 6A, the quality of BNIP3 blot needs to be improved.

6. PLOS authors have the option to publish the peer review history of their article (what does this mean?). If published, this will include your full peer review and any attached files.). If published, this will include your full peer review and any attached files.

.

Reviewer #1: **Yes:**Tao ZhangTao Zhang

Reviewer #2: No

---

## [Author Response · Author response to Decision Letter 1]

4 Feb 2026

Response to Editor's comments:

We sincerely thank the Editor and Reviewers for their constructive feedback on our manuscript (PONE-D-25-60616). We have carefully addressed all comments through extensive revisions, including restructuring of result section, statistical analysis, refined interpretation of the data presented that strengthens the mechanistic insight and therapeutic relevance of our findings. Corrections in the figures suggested by reviewers has been implemented and compiled according to the PLOS guidelines. We have significantly revised the manuscript and provided adequate responses to the reviewers' queries.

Response to Reviewers

Manuscript ID: PONE-D-25-60616

Thank you for the time and effort for providing valuable comments to improve our manuscript. We are grateful for your insightful suggestions, and we have incorporated the suggestions made by the reviewers. Those changes are highlighted as track changes in the manuscript. Please find below a point-by-point response to reviewers’ comments.

Reviewer #1:

This manuscript examines the effects of CoQ10 and melatonin on stress-induced dysfunction in human adipose-derived MSCs, and the authors conclude that antioxidant supplementation improves mitochondrial homeostasis, reduces cellular senescence, and enhances mitophagy, particularly through Parkin- and BNIP3-associated pathways.

Reviewer’s Comment -1. “While the topic is relevant, the manuscript in its current form is far from publication. The writing is difficult to follow, and the results section reads more like extended figure legends than a coherent scientific narrative. There is no logical flow in the results section and many statements are grammatically unclear, making the manuscript challenging to read.”

Author’s response: We sincerely thank the reviewer for this constructive feedback regarding the clarity and structure of our manuscript. We acknowledge that the original results section lacked coherent narrative flow and resembled extended figure legends, hindering readability.

To address it, we have re-written most of the results section by removing panel-by -panel descriptions and transformed into integrated paragraph statements where figures support the scientific narrative.

We have organized the result and discussion narrative progressing as follows: dose optimization of antioxidants and stress, treatment strategy validation, redox homeostasis - ROS accumulation and endogenous antioxidant defense mechanisms, mitochondrial membrane health and mitophagy protein expression.

Conducted line by line proof reading for spelling and grammatical mistakes.

Reviewer’s Comment -2. “Besides, the central conclusion—that antioxidants enhance clearance of damaged mitochondria—is not supported by the assays presented. JC-1, acridine orange staining, and changes in mitophagy-related proteins reflect alterations in mitochondrial membrane potential, lysosomal activity, and pathway signaling but do not demonstrate mitophagy flux, mitochondrial degradation, or mitochondria–lysosome colocalization. The data therefore do not establish mitochondrial clearance, and additional evidence is required to demonstrate the conclusion.”

Author’s response: We appreciate the reviewer for the clear critique you have provided us. It aptly points out the overinterpretation of mitophagy assays in our previously submitted manuscript. We completely agree that JC-1 is used for measuring the mitochondrial membrane potential (MMP), and the acridine orange staining image shows lysosomal accumulation. The mitophagy protein expression of our western blots indicate mitochondrial stress response priming. The above experiments do not confirm flux or clearance.

We have addressed this critical distinction through comprehensive revisions:

In our Previous Manuscript Correction made to our Revised Manuscript

1 Previous Manuscript: “CoQ10 and melatonin have improved clearance of damaged mitochondria by Parkin-mediated mitophagy when the stress is less and by BNIP3-mediated mitophagy with excess ROS”

Revised Manuscript: “CoQ10 and melatonin pretreatment to HADMSCs have modulated the expression of mitophagy receptors and LC3B in stressed HADMSCs”.

2 Previous Manuscript: “Optineurin protein expression shows active cellular response to stress, facilitating the clearance of damaged mitochondria through mitophagy.”

Revised Manuscript: “Optineurin protein levels are found to be reduced with antioxidant intervention shows a shift from optineurin dependent mitophagy to alternate pathways for explaining preserved mitochondrial health despite lower optineurin levels”

3 Previous Manuscript: “BNIP3 protein expression was significantly reduced in the stress group, whereas it was observed to be highly upregulated in the coQ10-pretreated stress group, indicating impaired mitochondrial clearance by mitophagy in stressed cells”.

Revised Manuscript: “BNIP3 protein expression was significantly reduced in the stress group, whereas it was observed to be highly upregulated in the coQ10-pretreated stress group, indicating activation of BNIP3 pathway in CoQ10 pretreatment group”.

4 Previous Manuscript: “The antioxidant-supplemented group and pretreated group have shown increased expression, indicating the clearance of damaged mitochondria”.

Revised Manuscript: “The antioxidant-supplemented group and pretreated group have shown increased expression, indicating the activation of BNIP3L/Nix mitophagy of damaged mitochondria.”

5 Previous Manuscript: “LC3B expression increased in all treated groups in comparison to the control, where the context of increased expression is different in each group, indicating that they have upregulated mitophagy, enhancing the clearance of damaged mitochondria, whereas in the stressed group there is no accumulation of NDP52, indicating the overwhelming damage causing impaired NDP52 and less optimal Parkin expression”.

Revised Manuscript: “LC3B expression increased in all treated groups in comparison to the control, where the context of increased expression is different in each group, indicating that they are, facilitating mitophagy, whereas in the stressed group there is no accumulation of NDP52, indicating the overwhelming damage causing impaired NDP52 and less optimal Parkin expression.”

Reviewer’s Comment -3. “Lastly, comparisons between pre- and post-treatment conditions are also not rigorously analyzed, while the author claimed pre-treatment worked better. The interpretation of the DPPH assay is similarly incomplete, particularly regarding the lack of improvement in antioxidant-pretreated stressed cells.”

Author’s response: We have addressed the comparison between pretreatment and post-treatment with statistical reanalysis and clarified interpretations. Pre-treatment significantly outperformed post-treatment across viability senescence (Two-way ANOVA P<0.0001). and senescence (Two-way ANOVA P<0.05).

We made the following addition in our revised manuscript:

For pre-treatment vs post treatment:

“In pretreatment settings where cells highest proliferation at 400nM concentration, achieving 117% & 98% of viability for CoQ10 & melatonin respectively. By contrast, in the post-treatment setting in which antioxidant treatments were given after the exposure of stress, the highest viability was at 200nM concentration, which is about 100.1% & 87.3% for coQ10 & melatonin respectively. Considering both the treatment strategies, it was observed that at 400nM concentration, pretreated cell viability was 117% and 98.8%, whereas post treated cell viability was 96.8% and 85.6%. These findings indicate that 400nM of coQ10 & melatonin pretreatment has been shown to improve HADMSC proliferation more effectively than post-treatment under stress conditions (P≤0.0001) (Two-way ANOVA). Analysis of staining intensity showed a greater reduction in senescent area with pre-treatment than with the post-treatment, with relative decrease of about 17.8% for CoQ10 and 46% for melatonin when comparing pre- vs. post-treatment conditions (Two-way ANOVA P<0.05).”

For DPPH assay Interpretation:

DPPH assay Interpretation earlier:

The increase in percentage of DPPH inhibition (Fig.3.C) in coQ10 and melatonin supplemented cells is about 31%(approx.) in both compared to the stressed group and can be correlated with greater radical-scavenging potential, which can be interpreted as enhancement of the cellular defense mechanism against oxidative damage by antioxidant supplementation. DPPH inhibition % in our HADMSCs—control, stressed cells, and pre-treated cells had no significant difference, and there was only a slight increase in antioxidant-supplemented cells, which might be secreted antioxidants or even the presence of supplemented antioxidants themselves. This further has to be strategically analyzed to arrive at a decision to conclude that antioxidant supplementation could benefit nearby cells due to the phenomenon of antioxidant secretion”.

DPPH assay Interpretation in our revised manuscript:

“DPPH assay of spent media showed that the antioxidant supplementation increased extracellular radical-scavenging activity, whereas subsequent stress exposure has reduced this effect. Media from CoQ10 and Melatonin-treated HADMSCs exhibited higher DPPH inhibition than serum-free medium (SFM), untreated control and Stressed (STR) groups, indicating enhanced release or it could be preservation of antioxidant capacity under basal conditions. In contrast, media from cells that were antioxidant-treated prior to stress (Q10+str) & (Mel+Str) showed intermediate DPPH inhibition, higher than the stressed group but lower than the antioxidant only group, suggesting that pretreatment partially maintains, but does not fully restore extracellular antioxidant activity in the presence of oxidative stress. DPPH inhibition increased by approximately 31% in antioxidant-pretreated groups for both compounds (Fig.3.C).DPPH assay of spent media revealed no significant differences in radical scavenging activity among control, stressed cells, and antioxidant pre-treated cells, with only modest increase in antioxidant-supplemented groups.

These finding suggest that secreted antioxidants or residual supplemented compounds contribute minimally to extracellular antioxidant capacity under the tested conditions.

This further has to be strategically analyzed to determine whether antioxidant supplementation enhance paracrine protection through secreted antioxidants”.

Reviewer’s Comment- 4. “Overall, the manuscript is far from publication-ready and requires extensive rewriting, clearer data interpretation, and more cautious conclusions. A full major revision followed by another round of evaluation will be necessary.”

Author’s response: We have systematically addressed through extensive major revisions. We have extensively rewritten the text, clarified our data interpretation, and moderated our conclusions to ensure they are fully supported by the evidence. Each specific concern has been addressed in the above point-by-point response, and we believe the revised manuscript is now significantly strengthened for publication.

Reviewer #2:

Vikraman and colleagues present a manuscript that examines the role of mitophagy in regulating senescence in human adipose-derived mesenchymal stem cells (HADMSCs) and the therapeutic potential of antioxidants melatonin and coenzyme Q10 (CoQ10). Stress-induced HADMSCs showed reduced mitochondrial membrane potential, increased ROS, and higher senescence-associated β-galactosidase activity. Antioxidant treatment reduced ROS and lipid peroxidation, restored mitophagy markers, and improved stem cell function. These findings suggest that antioxidant-induced mitophagy may reverse stem cell aging, enhance regeneration, and improve stem cell transplantation efficiency. Generally, the quality of the manuscript needs to be improved. And I have following comments:

1. The author should add page numbers and line numbers in the manuscript, so that the reviewers can clearly indicate which parts of the manuscript they are referring to.

Author’s response: Page number and line numbers added.

2. In “3. Results, 3.1”, the authors stated that “(Fig.1.A) Antioxidants showed maximum growth at 10-7 M, and elicited a cytotoxic response at 10-3 M concentration. Increased proliferative potential from 10-10 M to 10-7 M range.” The “(Fig.1.A)” should be put behind the sentences. The same applies elsewhere.

Author’s response: This correction has been implemented in the revised manuscript.

3. In Figure 1, the p values were labelled in a very confusing way. The authors seemed like only to choose some of the groups to label the p values without proper explanation.

Author’s response: The labelled p values have been explained in the result sections of the revised manuscript. Earlier P-value labelling in Figure 1 lacked transparency regarding statistical comparisons, this has been comprehensively resolved by explicit statistical reporting.

• Antioxidant treatment has shown a dose dependent effect on HADMSC proliferation and viability, where at lower concentrations antioxidants enhanced proliferation at 10-7 M (P<0.0001) (Fig.1.A)

• 100μM (P<0.0001) H₂O₂ exposure-maintained cell viability at around 83%, which was selected as an optimal condition to induce sublethal stress that directs cells towards senescence (Fig.1.C)

• LD50 for D-galactose in HADMSCs was determined to be 150mM (P<0.0001), where 100mM (P<0.0001) concentration was selected to induce cellular stress throughout the experiments (Fig.1.D).

• Overall, when comparing stressed versus antioxidant-treated HADMSCs prior to stress, there is significant viability (P<0.05) (Fig.1.F), both CoQ10 & melatonin exert a significant protective effect even at a concentration as low as 400nM.

• These findings indicate that 400nM of coQ10 & melatonin pretreatment has been shown to improve HADMSC proliferation more effectively (P<0.0001) (Fig.1.B)

4. In figure 2A and C, there is no scale bar. The labels of p value are not fully consistent with the manuscript.

Author’s response: We have added scale bars to the images- Figure 2A & Figure 2C. Statistical analysis were done again and P-values are corrected in our revised manuscript

5. In figure 3A, there is no scale bar. The way of showing inconsistent with previous figure.

Author’s response: the images in the Figure 3 is shown in 20x and the images in Figure 2 are in 10x. We have added scale bars using image J software.

6. In figure 5C, the image magnification is inconsistent among groups. And there is no scale bar.

Author’s response: With image J software we have added the scale bars for images in Figure 5C

7. In figure 6A, the quality of BNIP3 blot needs to be improved.

Author’s response: We are unable to conduct fresh gel electrophoresis because sample availability has run out. However we have repeated this blot several times , the results are consistent and reliable .

---

## [Decision Letter · Decision Letter 1]

23 Feb 2026

PONE-D-25-60616R1Synergistic Role of Mitophagy and Antioxidants in enhancing the fate and survival of Mesenchymal Stem Cells (MSCs) - Plausible role in Metabolic ReprogrammingPLOS One

Dear Dr. Dhanasekaran,

Thank you for submitting your manuscript to PLOS ONE. After careful consideration, we feel that it has merit but does not fully meet PLOS ONE’s publication criteria as it currently stands. Therefore, we invite you to submit a revised version of the manuscript that addresses the points raised during the review process.

The manuscript has been thoroughly revised and significantly improved. However, as noted by one of the reviewers, there are still a few crucial minor issues that must be addressed before the manuscript can be formally accepted for publication.

If applicable, we recommend that you deposit your laboratory protocols in protocols.io to enhance the reproducibility of your results. Protocols.io assigns your protocol its own identifier (DOI) so that it can be cited independently in the future. For instructions see: https://journals.plos.org/plosone/s/submission-guidelines#loc-laboratory-protocols. Additionally, PLOS ONE offers an option for publishing peer-reviewed Lab Protocol articles, which describe protocols hosted on protocols.io. Read more information on sharing protocols at . Additionally, PLOS ONE offers an option for publishing peer-reviewed Lab Protocol articles, which describe protocols hosted on protocols.io. Read more information on sharing protocols at https://plos.org/protocols?utm_medium=editorial-email&utm_source=authorletters&utm_campaign=protocols..

We look forward to receiving your revised manuscript.

Kind regards,

Hafiz Muhammad Umer Farooqi

Academic Editor

PLOS One

Journal Requirements:

Additional Editor Comments (if provided):

The manuscript has been thoroughly revised and significantly improved. However, as noted by one of the reviewers, there are still a few crucial minor issues that must be addressed before the manuscript can be formally accepted for publication.

Reviewers' comments:

Reviewer's Responses to Questions

**Comments to the Author**

1. If the authors have adequately addressed your comments raised in a previous round of review and you feel that this manuscript is now acceptable for publication, you may indicate that here to bypass the “Comments to the Author” section, enter your conflict of interest statement in the “Confidential to Editor” section, and submit your "Accept" recommendation.

Reviewer #1: All comments have been addressed

Reviewer #2: All comments have been addressed

2. Is the manuscript technically sound, and do the data support the conclusions?

Reviewer #1: Yes

Reviewer #2: Yes

3. Has the statistical analysis been performed appropriately and rigorously? 

Reviewer #1: Yes

Reviewer #2: Yes

4. Have the authors made all data underlying the findings in their manuscript fully available?

Reviewer #1: Yes

Reviewer #2: Yes

5. Is the manuscript presented in an intelligible fashion and written in standard English?

Reviewer #1: Yes

Reviewer #2: Yes

6. Review Comments to the Author

Reviewer #1: While the revised manuscript has improved scientifically, two major issues remain.

1. Title

The current title overstates the scope of the findings. Terms such as “synergistic role” and “metabolic reprogramming” imply mechanistic conclusions that are not directly supported by the presented data. The study demonstrates antioxidant-mediated protection and modulation of mitophagy-related protein expression, but does not provide direct evidence of mitophagy flux or metabolic reprogramming. The title should therefore be revised to more accurately reflect the experimental evidence.

2. Language and Clarity

The manuscript still requires substantial language revision. There are multiple grammatical inconsistencies and unclear constructions throughout the text. For example:

Missing conjunctions (e.g., “differentiate into multiple lineages, improve metabolic homeostasis”).

Incorrect or unnecessary verb usage (e.g., “Premature senescence in MSCs is caused…”).

Repeated sentence (e.g., "117% and 98.8%")

Inconsistent formatting of concentrations (e.g., 400Nm).

Abbreviations such as “Str” or “STR” not clearly defined at first mention.

P-values placed mid-sentence rather than at the end of statements.

A thorough professional language editing and careful proofreading are strongly recommended.

Reviewer #2: In this manuscript, authors showed that antioxidants (melatonin and CoQ10) restore mitophagy in stressed HADMSCs by reducing ROS and lipid peroxidation, enhancing mitophagy markers (Parkin, NDP52, BNIP3, Nix, LC3B), and reversing cellular senescence. This improves stem cell survival, metabolic function, and therapeutic potential for transplantation. The authors have addressed all my comments. I recommend to accept this manuscript.

7. PLOS authors have the option to publish the peer review history of their article (what does this mean?). If published, this will include your full peer review and any attached files.). If published, this will include your full peer review and any attached files.

.

Reviewer #1: **Yes:**Tao ZhangTao Zhang

Reviewer #2: No

---

## [Author Response · Author response to Decision Letter 2]

24 Mar 2026

Manuscript ID: PONE-D-25-60616R2

We sincerely appreciate the detailed feedback of the reviewers which had helped us to improvise our manuscript in this second revision. We have carried out the corrections as mentioned by the reviewer and journal requirements. We had re-viewed our references cited in our manuscript to ensure that none of the citations are made from retracted papers. We have cross-checked every reference with Google Scholar and Zotero to identify retracted status of any of the citations. There are no changes in the reference list. We have provided a point-by-point response to reviewer’s comments.

Reviewer #1:

While the revised manuscript has improved scientifically, two major issues remain.

Comment: 1. Title

The current title overstates the scope of the findings. Terms such as “synergistic role” and “metabolic reprogramming” imply mechanistic conclusions that are not directly supported by the presented data. The study demonstrates antioxidant-mediated protection and modulation of mitophagy-related protein expression, but does not provide direct evidence of mitophagy flux or metabolic reprogramming. The title should therefore be revised to more accurately reflect the experimental evidence.

Author’s Response:

In accordance to your suggestions and acknowledging the limitations of the presented data, we have changed the title and it is now titled as “Melatonin and Coenzyme Q10 mitigate Senescence in Human Adipose-Derived Mesenchymal Stem Cells by Restoring Mitophagy and Mitochondrial Proteostasis”.

Comment: 2. Language and Clarity

The manuscript still requires substantial language revision. There are multiple grammatical inconsistencies and unclear constructions throughout the text. For example:

• Missing conjunctions (e.g., “differentiate into multiple lineages, improve metabolic homeostasis”).

• Incorrect or unnecessary verb usage (e.g., “Premature senescence in MSCs is caused…”).

• Repeated sentence (e.g., "117% and 98.8%")

• Inconsistent formatting of concentrations (e.g., 400Nm).

• Abbreviations such as “Str” or “STR” not clearly defined at first mention.

• P-values placed mid-sentence rather than at the end of statements.

A thorough professional language editing and careful proofreading are strongly recommended.

Author’s Response:

Thank you for your detailed feedback regarding the language and clarity of manuscript. The examples mentioned, have been corrected in our manuscript. This helped us to fix the missing conjunctions and overall grammatical aspect of all the sentences.

• Missing conjunctions: (e.g., “differentiate into multiple lineages, improve metabolic homeostasis”- fixed in Page 2, line 23-25.

• Incorrect verb usage: (e.g., “Premature senescence in MSCs is caused…”- fixed in Page 3, line 3-5.

• Repeated sentence: 117% and 98.8%"- removed

• Inconsistent formatting of concentrations (e.g., 400Nm) has been corrected in Page 8, line 23.

• Abbreviations such as “Str” or “STR” has been corrected to “STR” throughout the manuscript and STR has been clearly mentioned as Stressed (STR) at first mention in page 10 and line 27.

• P-values initially placed mid-sentence rather than at the end of statements has been corrected and placed at the end of the statements, throughout the manuscript as per the journal style.

We have conducted line by line proof-reading to eliminate grammatical errors, confusing sentences or any inconsistencies like mentioned in the comments.

Reviewer #2: In this manuscript, authors showed that antioxidants (melatonin and CoQ10) restore mitophagy in stressed HADMSCs by reducing ROS and lipid peroxidation, enhancing mitophagy markers (Parkin, NDP52, BNIP3, Nix, LC3B), and reversing cellular senescence. This improves stem cell survival, metabolic function, and therapeutic potential for transplantation. The authors have addressed all my comments. I recommend to accept this manuscript.

Author’s Response:

Thank you for your time and effort for reviewing our manuscript. Your comments provided during the first revision has helped us immensely improve the quality of our manuscript. We are grateful for your insightful suggestions and thorough evaluation. Our heartfelt thanks for your positive assessment and your recommendation for the acceptance of our manuscript.

---

## [Decision Letter · Decision Letter 2]

8 Apr 2026

Melatonin and Coenzyme Q10 mitigate Senescence in Human Adipose-Derived Mesenchymal Stem Cells by Restoring Mitophagy and Mitochondrial Proteostasis

PONE-D-25-60616R2

Dear Dr. Dhanasekaran,

We’re pleased to inform you that your manuscript has been judged scientifically suitable for publication and will be formally accepted for publication once it meets all outstanding technical requirements.

An invoice will be generated when your article is formally accepted. Please note, if your institution has a publishing partnership with PLOS and your article meets the relevant criteria, all or part of your publication costs will be covered. Please make sure your user information is up-to-date by logging into Editorial Manager at Editorial Manager® and clicking the ‘Update My Information' link at the top of the page. For questions related to billing, please contact  and clicking the ‘Update My Information' link at the top of the page. For questions related to billing, please contact billing support..

Kind regards,

Hafiz Muhammad Umer Farooqi

Academic Editor

PLOS One

Additional Editor Comments (optional):

The authors have revised the manuscript according to the reviewers’ suggestions and have provided reasonable answers to their queries. In view of the above, the manuscript meets the publication criteria and scientific rigor of PLOS ONE and is therefore formally accepted for publication.

Reviewers' comments:

Reviewer's Responses to Questions

**Comments to the Author**

1. If the authors have adequately addressed your comments raised in a previous round of review and you feel that this manuscript is now acceptable for publication, you may indicate that here to bypass the “Comments to the Author” section, enter your conflict of interest statement in the “Confidential to Editor” section, and submit your "Accept" recommendation.

Reviewer #1: All comments have been addressed

2. Is the manuscript technically sound, and do the data support the conclusions?

Reviewer #1: Yes

3. Has the statistical analysis been performed appropriately and rigorously? 

Reviewer #1: Yes

4. Have the authors made all data underlying the findings in their manuscript fully available?

Reviewer #1: Yes

5. Is the manuscript presented in an intelligible fashion and written in standard English?

Reviewer #1: Yes

6. Review Comments to the Author

Reviewer #1: The authors have solved the concerns I have for the title and also improved the manuscript significantly. I suggest the acceptance of this paper.

7. PLOS authors have the option to publish the peer review history of their article (what does this mean?). If published, this will include your full peer review and any attached files.). If published, this will include your full peer review and any attached files.

.

Reviewer #1: **Yes:**Tao ZhangTao Zhang

---

## [Editor Report · Acceptance letter]

PONE-D-25-60616R2

PLOS One

Dear Dr. Dhanasekaran,

I'm pleased to inform you that your manuscript has been deemed suitable for publication in PLOS One. Congratulations! Your manuscript is now being handed over to our production team.

Kind regards,

on behalf of

Dr. Hafiz Muhammad Umer Farooqi

Academic Editor

PLOS One